# Synaptic and intrinsic mechanisms underlying development of cortical direction selectivity

**Arani Roy**[1,2], **Jason J Osik**[1,2], **Benyamin Meschede-Krasa**[1], **Wesley T Alford**[1], **Daniel P Leman**[1], **Stephen D Van Hooser**[1,2,3]*

[1]Department of Biology, Brandeis University, Waltham, United States; [2]Volen Center for Complex Systems, Brandeis University, Waltham, United States; [3]Sloan-Swartz Center for Theoretical Neurobiology Brandeis University, Waltham, United States

**Abstract** Modifications of synaptic inputs and cell-intrinsic properties both contribute to neuronal plasticity and development. To better understand these mechanisms, we undertook an intracellular analysis of the development of direction selectivity in the ferret visual cortex, which occurs rapidly over a few days after eye opening. We found strong evidence of developmental changes in linear spatiotemporal receptive fields of simple cells, implying alterations in circuit inputs. Further, this receptive field plasticity was accompanied by increases in near-spike-threshold excitability and input-output gain that resulted in dramatically increased spiking responses in the experienced state. Increases in subthreshold membrane responses induced by the receptive field plasticity and the increased input-output spiking gain were both necessary to explain the elevated firing rates in experienced ferrets. These results demonstrate that cortical direction selectivity develops through a combination of plasticity in inputs and in cell-intrinsic properties.

*For correspondence:
vanhoosr@brandeis.edu

**Competing interests:** The authors declare that no competing interests exist.

## Introduction

The physical world is represented in the primary sensory cortices through patterns of neuronal population activity evoked by sensory stimuli. Responses of individual neurons tuned to specific features of the sensory stimuli are essential building blocks of this population representation of the physical world. Feature-selective responses arise through a combination of precise patterns of neuronal connectivity and intrinsic cellular properties. A cortical neuron is an input-output device that receives thousands of cortical and sub-cortical synaptic inputs and in turn generates a spiking output, whose selectivity often differs dramatically from the selectivities of its sub-cortical inputs (*Hubel and Wiesel, 1962*; *Priebe and Ferster, 2012*). The cell-intrinsic properties gate this input-output transformation by controlling how the neuron summates inputs and then maps it on to a spiking output. While the patterns of synaptic inputs underlying some receptive field properties of cortical neurons have been uncovered in the mature cortex (*Jagadeesh et al., 1993*; *Reid and Alonso, 1995*), it remains unclear how the input patterns and the cell-intrinsic properties evolve during development (*Zhang and Linden, 2003*), and yet these features are a key component of understanding how cortical circuits compute in general.

In the mammalian primary visual cortex (V1), neuronal responses are particularly characterized by selectivity for two features of the visual stimuli: orientation (orientation selectivity) and direction of motion (direction selectivity) (*Hubel and Wiesel, 1959*). At the time of eye opening in carnivores and primates, while neurons already exhibit orientation selectivity, they respond about equally to motion in the two opposite directions orthogonal to the preferred axis of orientation. Selectivity to direction of motion develops rapidly in the days and weeks following the onset of visual experience

(*Hatta et al., 1998*; *Li et al., 2006*; *Clemens et al., 2012*; *Smith et al., 2015*), and can be accelerated in the lab through exposure to moving stimuli (*Li et al., 2008*; *Van Hooser et al., 2012*). In cats and primates, it has been demonstrated that a set of inputs activating the cell at varying spatial positions and temporal latencies following a specific spatiotemporal order underlies direction selectivity in the visually mature state (*DeAngelis et al., 1993b*; *Jagadeesh et al., 1993*; *McLean et al., 1994*; *Livingstone, 1998*). But the mechanisms underlying the development of this precise pattern of spatiotemporal activation, and consequently of direction selectivity, are poorly understood. Additionally, the contributions of cell-intrinsic properties to the development of direction-selectivity also remain largely unexplored.

Therefore, to clarify the contributions of synaptic and cell-intrinsic properties to the development of direction selectivity, we carried out in vivo intracellular recordings in V1 from two groups of ferrets at different stages of visual development: visually naive (post-natal day P30-34) and visually experienced (P40-60). Using reverse correlation of membrane potential (Vm) responses to sparse noise stimuli, we constructed the spatiotemporal receptive fields (STRF) of neurons in the two groups and compared them. We found significant differences in the spatiotemporal organization of receptive fields between the visually naive and experienced groups. Surprisingly, the enhancement in Vm responses brought about by these receptive field rearrangements did not successfully predict the developmental enhancement in spiking responses. We tested whether a concomitant developmental alteration in intrinsic excitability, which gates the input-output transformation of neurons, could underlie this discrepancy. Consistent with this idea, we found that V1 neurons in visually experienced animals had stronger gains in Vm-to-spike transformation, and that this gain enhancement was accompanied by a lowering of spike threshold. Changes in subthreshold responses and voltage-to-firing rate gain enhancement both had to be considered to explain the total increases in firing rates that are observed over this period (*Clemens et al., 2012*), signifying that a combination of synaptic and intrinsic plasticity mechanisms underlie development of direction selectivity.

## Results

At the time of eye opening, around 30–33 days of postnatal age (P30-P33), neurons in ferret visual cortex exhibit robust selectivity for stimulus orientation but only very weak selectivity for stimulus direction (*Li et al., 2006*; *Li et al., 2008*). Selectivity for stimulus direction emerges over the weeks following eye opening through a process that requires visual experience (*Li et al., 2006*). In cortical simple cells in the visually experienced state, direction selectivity is thought to arise from spatiotemporally selective inputs activating the cell at varying spatial positions and temporal latencies (*DeAngelis et al., 1993b*; *Jagadeesh et al., 1993*; *McLean et al., 1994*; *Livingstone, 1998*). According to this model (*Reichardt and Poggio, 1976*; *Reichardt, 1987*), a direction-selective neuron is maximally activated when its inputs selective for specific spatial locations within the receptive field are activated by a moving stimulus in a specific temporal order – with the inputs with the longest latencies activated first and inputs with progressively shorter latencies sequentially activated thereafter. This spatiotemporally sequenced order of activation ensures that the subthreshold inputs activated by the preferred direction of motion arrive at the soma simultaneously and achieve maximal summation, leading to a strong response. In contrast, a stimulus moving in the null direction activates the shortest latencies first followed by progressively longer latencies, thereby leading to suboptimal summation of activity, and consequently lower responses in the postsynaptic neuron. A direction-selective simple cell with its inputs organized in this manner exhibits a stereotypical 'slant' in its spatiotemporal receptive field.

In order to examine the initial state of spatiotemporal receptive fields (*Figure 1*) and to understand how these fields are altered during the short critical period for the emergence of direction selectivity, we carried out in vivo intracellular recordings in the visual cortex of anesthetized ferrets using sharp microelectrodes. Intracellular recordings were used because naive visual cortical neurons exhibit lower firing rates than experienced animals (*Clemens et al., 2012*; *Popović et al., 2018*), and we wanted to be able to examine the receptive field properties of the subthreshold voltage in addition to spiking. Experimental animals were split in to two age groups: visually naive ('naive': age P30-34, n = 10 animals, 23 cells) and visually experienced ('experienced': age P40-60, n = 11 animals, 29 cells). Drifting sinusoidal grating stimuli were used to measure responses to stimulus orientation and direction. Cells were classified as simple or complex by comparing the fundamental (F1)

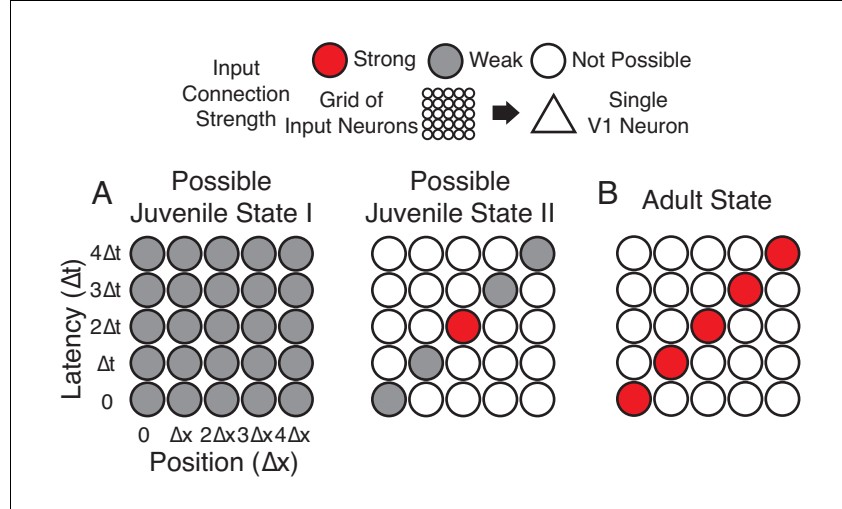

**Figure 1.** Hypotheses of initial naive states. (**A**) In Possible Juvenile State I, receptive fields might be initially broad and weak in space and time and later sharpened to exhibit a slant. In this state, the lack of direction selectivity comes from the symmetry in the broad initial receptive field. In Possible Juvenile State II, there is a compact initial input that is constrained to grow only at certain positions and latencies, either by addition of new synaptic inputs or by changes in the properties of existing inputs. In this state, the lack of direction selectivity comes from the symmetry of compact strong input, and the weak inputs do not contribute at the naive stage. (**B**) Experienced state, with a characteristic directional slant in space time, indicating that the cell would respond to a leftward moving stimulus at an appropriate velocity.

and the DC components (F0) of the visual responses (*Carandini and Ferster, 2000*; *Mechler and Ringach, 2002*; *Priebe et al., 2004*). We first converted the Vm and spike responses of each recorded neuron to the frequency domain using Fast Fourier Transform (FFT) and then analyzed all response properties for the F0 and F1 components separately as well as combined (F0+F1). We calculated Vm and spike modulation ratios by dividing the amplitude of the F1 component by that of the F0 component of the respective responses. Cells with spike modulation ratio equal to or greater than 1.0 were classified as simple, while the rest were categorized as complex (*Figure 2—figure supplement 1*). Neurons that did not fire sufficient spikes were classified as simple if their Vm modulation ratio was greater than 0.5. According to this classification, our dataset contained 10 simple and 13 complex cells in the naive group, and 16 simple and 13 complex cells in the experienced group. After a cell's orientation and direction preferences were identified by the responses to grating stimuli, sparse noise bar stimuli were shown to examine the spatiotemporal receptive fields only for the simple cells, because such analysis of linear summation properties apply only to simple cells.

## Emergence of enhanced direction selectivity following visual experience

We constructed orientation and direction tuning curves from recorded Vm and spiking activities in response to sinusoidal gratings. *Figure 2AB* shows an example simple cell each from a naive and an experienced ferret. The neuron from the naive ferret (*Figure 2AB*, left panel) responded strongly to horizontal gratings, irrespective of whether they were moving upward or downward, while showing only weak subthreshold responses to vertical gratings. In contrast, the neuron from the experienced ferret (*Figure 2AB*, right panel) strongly responded only to vertical gratings drifting to the right while responding minimally to all other stimuli. Direction selectivity index values for spiking activity and Vm were much larger for the experienced example cell than for the naive cell. On average, the experienced cells exhibited stronger direction selectivity compared to the naive cells, both at the subthreshold Vm and the suprathreshold spiking levels (*Figure 2C*, Direction selectivity index or DSI. Top, Vm: naive: mean = 0.16 ± 0.02, n = 23; experienced: mean = 0.35 ± 0.04, n = 29; p<0.001, WRS test; bottom, spike: naive: mean = 0.3 ± 0.05, n = 14 (9 cells did not spike); experienced: mean = 0.66 ± 0.05, n = 29; p<0.001, WRS test).

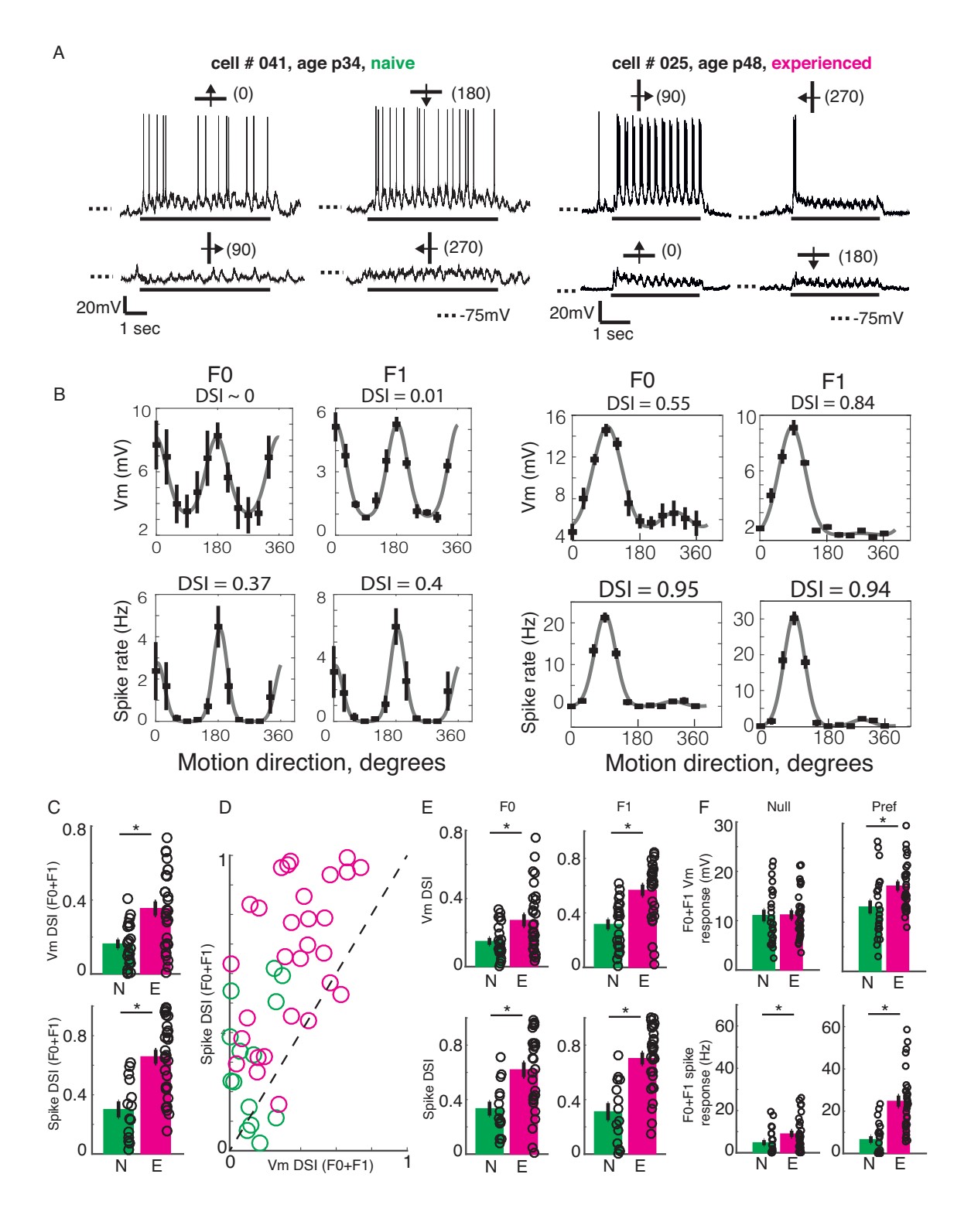

**Figure 2.** Developmental increase in direction selectivity of membrane potential (Vm) and spiking responses in V1 cells. (**A**) Vm responses of a simple cell from a visually naive and an experienced ferret each. For each cell, single-trial responses to presentation of sinusoidal gratings moving in both directions orthogonal to the preferred orientation (top) and the non-preferred orientation (bottom) are shown. The stimulus orientations and motion directions are indicated above each trace with a solid bar and arrow, respectively, and the angles representing the specific directions of motion are

*Figure 2 continued on next page*

*Figure 2 continued*

given in parentheses. Solid bar below each trace indicates the duration of the stimulus display and dotted lines indicate the Vm level of −75 mV. For both cells, the gratings drifted at a temporal frequency of 4 Hz and the Vm oscillated at the same frequency with a strong response to each grating cycle. (B) Direction tuning curves of the cells in A, plotted separately for Vm (top) and spiking (bottom) responses, and for the F0 or DC (left) and the F1 or fundamental (right) components of the responses. Direction selectivity index (DSI) value calculated from each tuning curve is given above each curve. Both F0 and F1 DSI values are higher in the cell from the experienced animal. (C) Mean total Vm and spike DSI values, calculated from F0+F1 responses, compared between naive (N, green) and experienced (E, magenta) animal groups. (D) Vm and spike total DSI (F0+F1) values for each cell plotted against each other. Dashed line denotes the line of unity. (E) Same as C, but the DSI values are calculated separately for the F0 and the F1 components of response. (F) Mean total (F0+F1) Vm and spiking responses to the null and preferred direction of motion compared between naive and experienced groups. For all panels: error bars denote SEM; circles denote values for individual cells; asterisks denote significant differences at p<0.05 level, Wilcoxon rank sum test.

The online version of this article includes the following figure supplement(s) for figure 2:

**Figure supplement 1.** Scatterplot of spike and Vm modulation ratios (F1/F0) for allcells recorded.

In a majority of cells, the DSI for spiking responses were greater than the DSI for Vm responses. A pairwise comparison of Vm and spike DSI within each neuron revealed that the spike DSI was significantly higher than Vm DSI in both the naive and the experienced groups (*Figure 2D*; naive, p=0.001, n = 14; experienced, p<0.001, n = 29; signed rank test). Thus, in both groups, the outputs of V1 cells were more sharply tuned for direction than their summed inputs. Also, to test if the developmental enhancement of DSI in V1 cells were restricted to the mean (F0) or the modulated (F1) component or affected both, we calculated DSI separately for the F0 and the F1 response components. F0 and F1 DSI for both sub- and suprathreshold responses in the experienced group were significantly higher than the naive group (*Figure 2E*; F0: Vm: naive: mean = 0.15 +/- 0.02, n = 23; experienced: mean = 0.27 +/- 0.04, n = 29; p=0.02, WRS test; spike: naive: mean = 0.33 +/- 0.05, n = 14; experienced: mean = 0.62 +/- 0.05, n = 29; p=0.003, WRS test; F1: Vm: naive: mean = 0.32 +/- 0.04, n = 14; experienced: mean = 0.57 +/- 0.04, n = 29; p<0.001, WRS test; spike: naive: mean = 0.31 +/- 0.06, n = 14; experienced: mean = 0.7 +/- 0.04, n = 29; p<0.001, WRS test), demonstrating that both the mean and the modulated component of visual responses became more direction-selective during development.

An increase in direction selectivity index values could be realized either by an increase in responses to the preferred direction, a decrease in responses to the null direction, or both. To distinguish between these possibilities, we analyzed the Vm and spiking response amplitudes to the preferred and the null direction stimuli for each cell. The total (F0 plus F1) responses to the preferred and null stimuli were compared between the naive and experienced groups, separately for Vm and spiking responses (*Figure 2F*). At the level of Vm (*Figure 2F*, top), the null responses (left) did not show any significant change over development, whereas the preferred responses (right) showed a modest but significant increase from the naive to the experienced group (Null, naive: mean = 11 ± 1.2, n = 23; experienced: mean = 11.2 ± 0.9, n = 29, p=0.8, WRS test; Preferred, naive: mean = 13.1 ± 1.3, n = 23; experienced: mean = 17.3 ± 0.9, n = 29, p=0.008, WRS test). At the level of spiking (*Figure 2F*, bottom), both the null and the preferred responses underwent developmental enhancement (null, naive: mean = 4.8 ± 1.2, n = 23; experienced: mean = 9 ± 1.5, n = 29, p=0.01, WRS test; preferred, naive: mean = 6.6 ± 1.6, n = 23; experienced: mean = 24.7 ± 2.6, n = 29, p<0.001, WRS test). Therefore, the increase in direction selectivity following visual experience was driven primarily by a robust increase in responses to the preferred direction of motion and not by a decrease in response to the null direction of motion. Notably, while this increase in response to the preferred direction was modest (32%) at the Vm level, it was dramatic for spiking activity (274%).

## Reorganization of simple cell spatiotemporal receptive fields following visual experience

The increase in Vm responses to the preferred direction raises the question as to what changes in the spatiotemporal receptive field configurations might underlie these increases. To compare spatiotemporal receptive field structures of the Vm responses between simple cells from the naive and experienced groups, we performed reverse correlation of the response of the Vm to sparse, one-dimensional noise visual stimulation (*Reid et al., 1987*; *Reid et al., 1991*; *Reid et al., 1997*; *Ringach et al., 1997*; *Priebe and Ferster, 2005*; *Rust et al., 2005*; *van Kleef et al., 2010*). The

noise stimulus consisted of black, white and gray bars, angled at the cell's preferred orientation, that changed pattern every 100 ms (*Figure 3A*). Positive cross-correlation values were obtained if a white bar (contrast 1) led to increased Vm or a black bar (contrast −1) led to decreased Vm (ON subunit), and negative cross-correlation values were obtained if a white bar led to decreased Vm or a black bar led to increased Vm (OFF subunit). For every spatial location, the cross-correlation values were plotted at varying lag times, thereby allowing assessment of the latencies at which the high or low cross-correlation values were obtained. Because visually driven spiking activity was weak in naive ferrets, we focused our analysis primarily on the Vm responses, for which a reliable spatiotemporal receptive field could be computed for every simple cell tested.

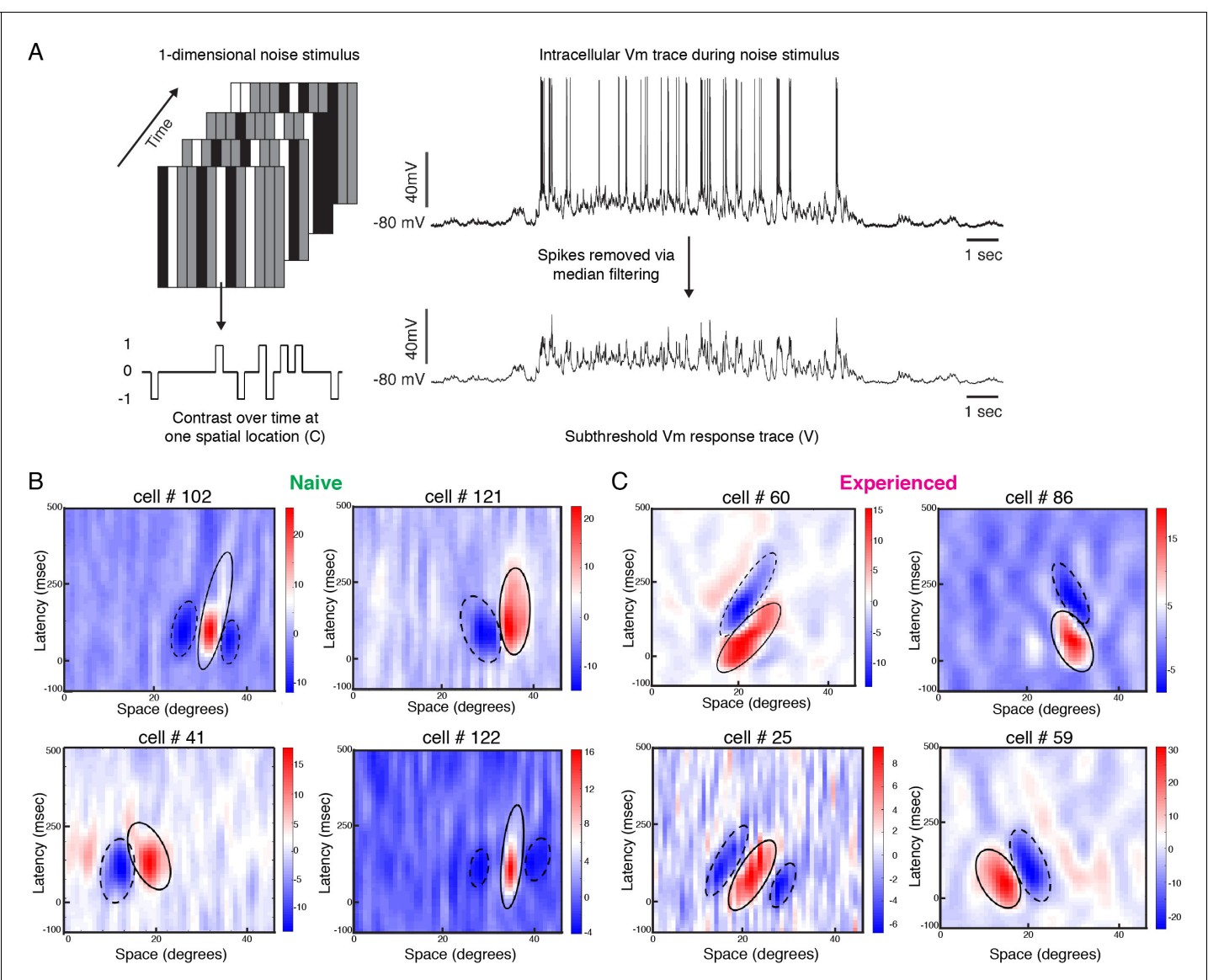

**Figure 3.** Developmental reorganization of linear spatiotemporal receptive fields in visual cortical simple cells. (A) Schematic of sparse noise stimulus, rotated to match each cell's preferred orientation. Spikes were removed and the membrane potential was correlated with the stimulus at each position. (B) Vm spatiotemporal receptive fields of four simple cells each from naive animals. X-axis represents spatial location, y-axis represents latency from onset of visual stimuli, and the cross-correlation coefficient between the stimulus contrast and Vm values are represented by the color. Black lines outline the ellipses fitted to the ON (continuous) and OFF (dashed) subunits. Raw correlations are shown, which are blurred by the time step of the stimulus (100 ms), so some ellipses overlap zero. (C) Same, for four cells from experienced animals. Receptive fields in experienced animals appeared more elongated in space time, and more slanted, consistent with the increased direction selectivity that was observed.

*Figure 3B, C* shows Vm spatiotemporal receptive fields of four cells recorded from a naive ferret and four cells recorded from an experienced ferret. Consistent with previous findings (*Reid et al., 1987*; *Reid et al., 1991*; *DeAngelis et al., 1993b*; *Priebe and Ferster, 2005*), the spatiotemporal receptive field subunits from direction-selective neurons in experienced animals were slanted in space-time. For cell #60, gratings moving from right to left on the monitor would activate the inputs with longer latencies first, progressively moving to shorter latencies as the grating moves across the visual field, thereby leading to maximal response. Stimuli moving from left to right, on the other hand, will activate the shorter latency inputs followed by the longer latency inputs, thereby leading to sub-optimal responses. The spatiotemporal receptive field of Cell #86 was also slanted, albeit in the opposite direction.

The spatiotemporal receptive fields from the neurons in naive animals were not completely unstructured, and exhibited several properties of neurons from experienced animals such as alternating ON and OFF subunits and a small slant in space-time. However, the subunits from the experienced cells were longer and narrower in profile, and relatively more slanted in space-time.

To rigorously quantify the degree to which subunit structure differed between naive and experienced animals, we fitted ellipses to the individual ON and OFF subunits (*Leong et al., 2016*) and compared several shape parameters of these ellipses (see **Materials and methods**). We chose a set of nine parameters that adequately captured the relevant spatial and temporal characteristics of the receptive fields (*Figure 4A*) – 'eccentricity': eccentricity of the fitted ellipse; 'major axis': the length of the major axis of the fitted ellipse; 'minor axis': the length of the minor axis of the fitted ellipse; 'orientation': the angle the ellipse subtended on the space (x-) axis; 'area': the full area covered by the ellipse, in pixels; 'spatial extent': projection of the ellipse on the space axis; 'temporal extent': projection of the ellipse on the time/latency axis; 'minimum latency': the lower bound of the fitted ellipse on the time axis signifying the shortest latency response; and 'maximum latency': the upper bound of the fitted ellipse on the time axis signifying the longest latency response. Wilcoxon rank sum test comparisons revealed that six out of these nine parameters were significantly different between the naive and the experienced groups: eccentricity, major axis, minor axis, orientation, temporal extent and minimum latency. The shape parameter that is most relevant to direction selectivity is the eccentricity of the ellipses describing the subunits. If subunits are broad along the minor axis of the ellipse, they will enable some response summation from stimuli moving in either direction, thus leading to weaker direction selectivity. Also, if subunits are short along the major axis of the ellipse, there might not exist sufficient latency differences to facilitate differential input summation. Therefore, to optimally support high direction selectivity, receptive field subunits would have to be narrow on the minor axis and elongated on the major axis, which would result in high eccentricity.

Consistent with this idea, we found that the eccentricity measured from the experienced subunits were significantly higher (p<0.001), and was caused by both a longer major axis (p<0.03) and a shorter minor axis (p<0.03, *Figure 4A*). Also, the subunits from the experienced animals subtended a smaller angle on the space axis (orientation: p<0.005), which would indicate that the cells from experienced animals are tuned to higher motion velocities, consistent with data from actual V1 recordings in cats (*DeAngelis et al., 1993a*). The overall area (p<0.6) and the spatial projection (p<0.35) of the subunits did not significantly differ. However, the projection on the latency axis was significantly longer in the experienced group (p<0.04), suggesting that a longer range of response latencies became available in the experienced animals. This extension of the latency range was achieved solely by stretching the receptive field subunits towards the lower latency side: while the maximum latencies of the subunits were not significantly different (p<0.3), the minimum latencies were significantly lower in the experienced group (p<0.05). Consistent with this data, we also found that the Vm response latency distributions were significantly shifted towards shorter latencies in the experienced animals (*Figure 4E*), and these results mirror reduced latencies observed in experienced vs. naive neurons in ferret lateral geniculate nucleus (*Tavazoie and Reid, 2000*).

If the changes in Vm spatiotemporal receptive field structure resulted in improved discrimination between the preferred and null directions of motion, then the receptive field structure parameters should correlate with Vm direction index measured in each cell. Therefore, we computed linear correlations between cell-averaged shape parameters and the cells' Vm direction index values (*Figure 4B*). We found that the Vm direction index of the cells increased with increasing 'narrowness' of the subunits (longer 'major axis', shorter 'minor axis' and higher 'eccentricity'), 'orientation', if the one low outlier is removed, and decreasing minimum latencies accessible (lower 'min latency'). Other

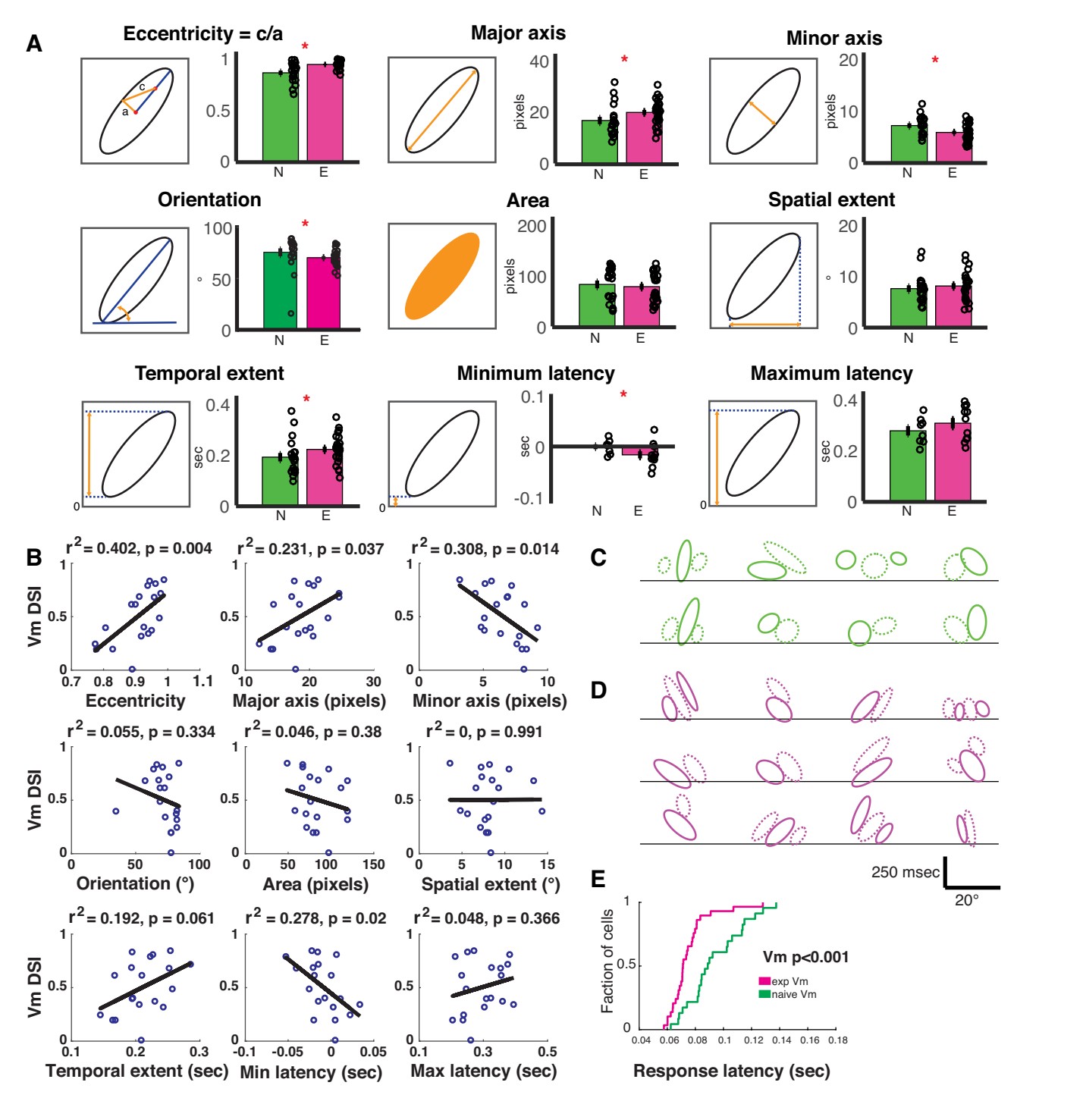

**Figure 4.** Spatiotemporal receptive fields become more extended, slanted in space-time with short-term visual experience. (**A**) Quantification of the nine parameters defining the characteristics of the spatiotemporal receptive field subunits. In each panel, the parameter being quantified is described by a schematic on the left, and on the right is a bar plot showing the mean ± SEM of the parameter values in the naive (N, green, 19 subunits from 8 cells) and experienced (E, purple, 26 subunits from 12 cells) animals. Black circles denote individual subunit values. Red stars denote statistical significance at p<0.05 level via WRS test. (**B**) Relationship between cell-average spatiotemporal receptive field structure parameters and Vm DSI for all simple cells. The R[2] and p values for each linear correlation are shown above each plot. Raw correlations were used in the calculations, which are blurred by the time step of the stimulus (100 ms), so some ellipses begin earlier than latency 0. If the outlier (orientation <45 ˚) in orientation is removed, then the r[2] is 0.26 and p≤0.03. (**C**) All subunits (ON = solid, OFF = dashed) for naive cells. Lines are at 0 time lag. Scale bar shows 250 msec

*Figure 4 continued on next page*

*Figure 4 continued*

by 20° of visual angle. Note that correlations are raw, so they are blurred 100 ms by the stimulus timesteps (some subunits begin before time 0). (D) Same, for experienced cells. Note the earlier onset responses and the greater slants. (E) Cumulative histogram of response latencies of naive and experienced simple cells to grating stimuli. Latency is defined as the first time when the voltage signal reaches six standard deviations.

The online version of this article includes the following figure supplement(s) for figure 4:

**Figure supplement 1.** Predicting Vm DSI of the F1 component from the simple cell STRFs.

parameters did not show significant correlation. These data further corroborate the idea that narrower spatiotemporal input profiles and availability of shorter latency inputs exert a strong impact on directional summation of inputs in simple cells of experienced animals, thereby leading to higher direction selectivity. Additionally, the Vm DSI predicted from the STRFs, by converting the STRFs into the frequency domain via 2-dimensional FFT, showed a linear correlation with the actually measured Vm DSI of the F1 components of the simple cells (*Figure 4—figure supplement 1*), further demonstrating that the STRF structures influenced the direction selectivity index values of the cells. The subunit shapes for all cells are shown in *Figure 4CD*.

In sum, following visual experience, the Vm receptive fields exhibited marked reorganization involving increased eccentricity of the space-time profile and a shift of response latencies towards lower values, leading to improved discrimination of directional inputs at subthreshold level and thus a more direction-selective Vm response.

## Plasticity of cellular intrinsic properties following visual experience

We saw a sharp rise in visually evoked spiking responses in the experienced animals. The fraction of cells that fired action potentials in response to drifting grating stimuli was significantly higher in the experienced group (simple: naive, 7/10; experienced, 16/16; p<0.05, complex: naive, 7/13; experienced, 13/13; p<0.05, Fisher Exact test). Thus, every recorded cell in the experienced group fired action potentials, but 9/23 cells in the naive group did not fire any action potentials at all. However, enhancement of Vm responses mediated by reorganization of STRF structure alone could not account for this enhanced spiking. As we saw in *Figure 2F*, the responses to the preferred direction underwent modest developmental increase (32%) at the Vm level, but the increase was dramatic for spiking activity (274%). Therefore, we argued that changes in cell-intrinsic properties also must have played a role in amplifying the effects at the Vm level. Visual inspection of spike waveforms revealed a couple of striking differences between the naive and the experienced groups: in the naive group, the spikes were significantly broader and the take-off point of the action potential (spike threshold) was higher (*Figure 5ABC*). These observations lead us to examine cell-intrinsic properties more carefully.

To characterize changes in cell-intrinsic properties, we examined several parameters of the action potential. These parameters included the maximum rate of change in Vm preceding a spike (max dV/dt), which has been shown to correlate with the density of sodium channels (*Colbert and Pan, 2002*; *Kole et al., 2008*; *Hu et al., 2009*; *Grubb and Burrone, 2010*; *Kuba et al., 2010*); a quantity called the spike kink (one way to define threshold), which is based on the time when the voltage obtains a sufficiently large rate-of-change relative to its maximum (*Azouz and Gray, 1999*; and the full width at half maximum of the spike waveform.

In vivo, the maximum dV/dt exhibited a strong increase between naive and experienced animals (*Figure 5C*, naive mean: 172.2 ± 64.6 mV/ms, experienced mean: 253 ± 78.3 mV/ms, WRS test p<2.0e-04), a small empirical drop in spike kink voltage (*Figure 5D*, naive mean: 21 ± 0.4 mV, experienced mean: 19.1 ± 4.7 mV, WRS test p=0.21, not statistically significant), and a very substantial decrease in action potential width (*Figure 5E*, naive mean: 1.6 ± 0.7 ms, experienced mean: 1.0 ± 0.4 msec, WRS test p<4.3e-04).

These in vivo measurements of action potential properties are altered by the ongoing activity of the cortex, so we also made these measurements in synaptic transmission-blocked visual cortical slices prepared from naive and experienced ferrets. Maximum dV/dt measurements in slice (*Figure 5F*) strongly resembled those measured in vivo (naive mean: 162.1 ± 51.1 mV/ms, experienced mean: 264.4 ± 61.9 mV/ms, WRS n vs. e, p<2.4e-05). The spike kink voltage exhibited a small, significant decrease with experience (naive mean: −25 ± 3.5 mV, experienced mean: −30.2 ± 3.7

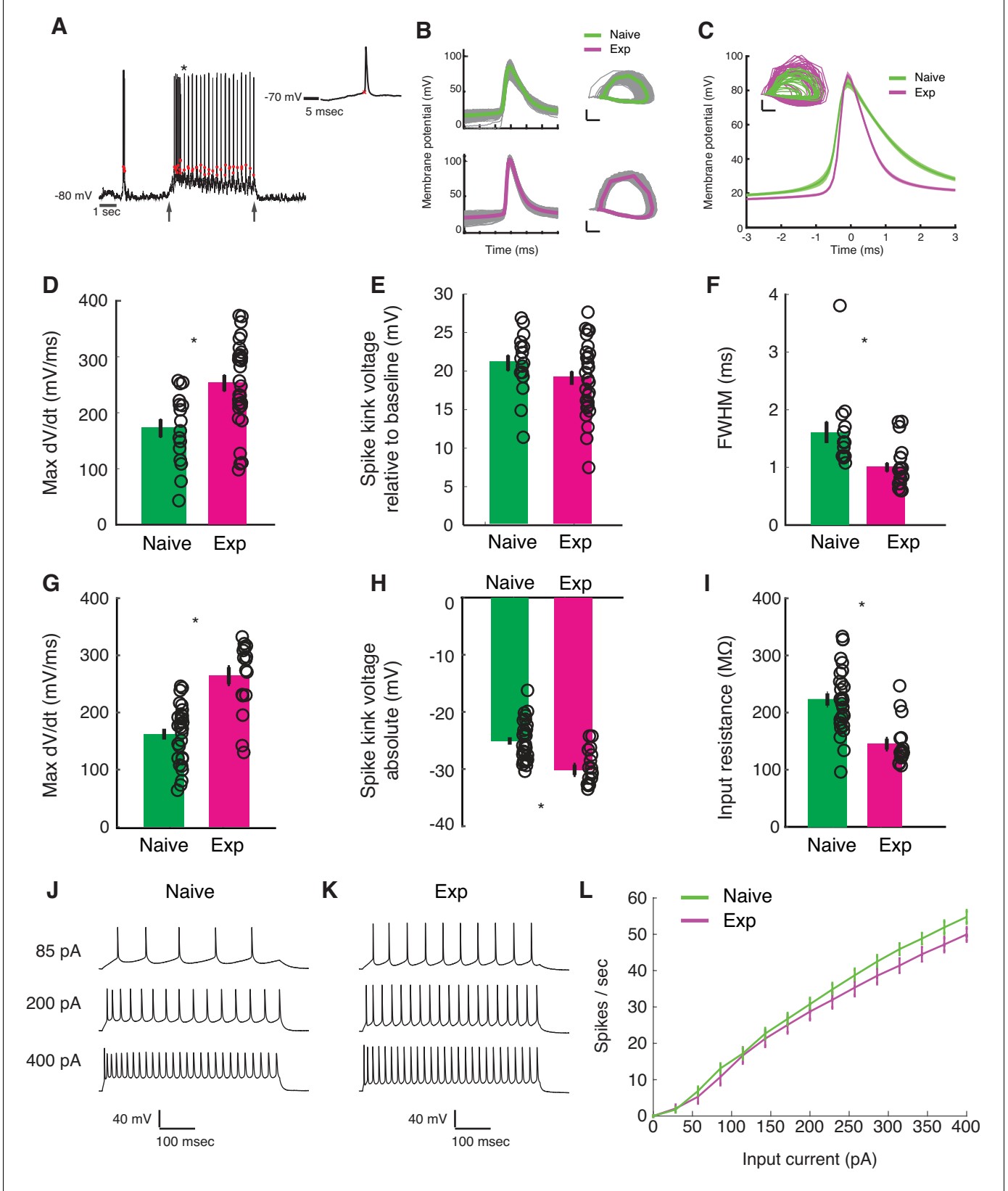

**Figure 5.** Developmental enhancement of excitability in cortical cells. (**A**) Detection of spike kink voltage from single action potentials in vivo (***Azouz and Gray, 1999***). (**B**) Action potentials and spike phase diagram from an example naive (top) and experienced cell (bottom). Individual spike

*Figure 5 continued on next page*

*Figure 5 continued*

traces shown in gray. In spike phase diagram, membrane potential is on the horizontal axis (scale bar is 20 mV) and dV/dt is on the vertical axis (scale bar is 100 mV/msec). (C) Average and standard error of the mean of voltage waveforms for naive (N = 14) and experienced (N = 29) cells, showing much wider action potentials in naive animals. Shaded area is standard error of the mean across cells. Inset: phase diagram of mean spike waveforms; membrane potential is on the horizontal axis (scale bar is 20 mV) and dV/dt is on the vertical axis (scale bar is 100 mV/msec). (D) Maximum dV/dt for naive and experienced cells in vivo (WRS test, p<2.0e-04); (E) spike kink voltages in vivo (WRS test, p=0.2); (F) action potential full width at half height (WRS test, p<4.3e-04) in vivo. (G) Maximum dV/dt for synaptically isolated cells in brain slices for naive and experienced animals. Ex vivo measurements showed similar maximum dV/dt values as in vivo measurements (N = 29 naive, N = 17 experienced, WRS, p<2.4e-05), showing similar values as in vivo measurements. (H) Spike kink voltage exhibited a significant decrease when measured in isolated cells from visually experienced animals (WRS p<1.6e-04). (I) Spike kink reductions and increased maximum dV/dt co-occurred with developmentally typical decreases in membrane resistance (WRS, p<1.2e-04), suggesting that experienced cells would take longer to charge from rest even though they spike at lower voltages. Example firing rate vs. current curves for a cell from a naive (J) animal and an experienced (K) animal are shown, and average firing rate vs. current curves are shown (L). The increased excitability with age and the decreased input resistance produced firing rate vs. current relationships that were not different between the two cases: (Kruskal-Wallis test of slopes: p<0.068). For all panels, p-values as listed are before Bonferroni correction; significant * indicates significant difference after Bonferroni correction (which did not change results for these tests).

mV), consistent with the idea that the experienced cells were more excitable near threshold. As expected for developing animals, input resistance exhibited a drop with developmental age (naive mean: 222.9 ± 57.9 MΩ, experienced mean: 144.8 ± 39.3 MΩ), meaning that it took more time to charge older cells from rest than naive cells. The firing rate vs. input current curves (F-I curves) were not different across the two groups (*Figure 5IJK*; f-I curve Kruskal-Wallis test of linear slopes: p<0.068).

## Changes in the voltage-to-firing rate transform contribute to development of visual responses

Although we found evidence for changes in excitability with development, the preceding analyses could not tell us quantitatively how these changes might impact sensory responses. To address this, we examined the voltage-to-firing rate transformation following methods of Priebe and Ferster (*Priebe et al., 2004*; *Priebe and Ferster, 2005*; *Priebe and Ferster, 2006*). In brief, voltage waveforms were median-filtered to remove spikes (*Figure 6AB*), and both spike trains and spike-trimmed voltage waveforms were binned into 30 ms intervals (*Figure 6C*) to generate pairs of Vm and firing rate observations. Voltage-to-firing rate relationships, with sliding 1 mV empirical mean curves and power law fits to the data, are shown for four examples cells in *Figure 6DE*. Empirical mean voltage-to-firing rate curves for all naive and experienced cells are plotted in *Figure 6F*, and individual voltage-to-firing rate curves for all cells are shown in *Figure 6—figure supplement 1*. Generally, most naive cells exhibited smaller firing rates for a given voltage than the experienced cells, which reflects the increased excitability of the mature neurons. On average, experienced cells exhibited significantly higher firing rates than naive cells over a wide range of voltages (*Figure 6G*).

To calculate the average impact of changes in selectivity and firing rates due to membrane voltage and the voltage-to-firing rate transform across naive and experienced cortical neurons, we modeled the results phenomenologically by passing the actual voltage waveforms that were recorded for each stimulus through each cell's power law fit function to generate firing rate responses on a bin-by-bin basis. This phenomenological model did an excellent job of reproducing the actual firing rates recorded for each stimulus and each cell for both naive and experienced cells (*Figure 7AB*). To explore the impact of changes in membrane voltage and voltage-to-firing rate transforms separately on the direction index values (*Figure 2*, *Figure 7C*) and firing rates (*Figure 7E*) we observed, we created 10,000 bootstrap simulations (*Press et al., 1993*) of experiments with 20 cells that used the waveforms and voltage-to-firing rate transform for different cell groups. In each simulation, a cell was chosen at random (with replacement) to provide the response Vm waveforms, and another cell was chosen at random (with replacement) to provide the voltage-to-firing rate transform. This rough phenomenological model assumes that voltage responses and gains can be selected independently; in the actual data, these quantities are not completely independent and the random mixture model slightly overestimates actual firing rates from the population. When membrane voltages and voltage-to-firing rate transforms were both selected from naive cells, the simulations yielded relatively low direction selectivity index values (*Figure 7D*) and low firing rate responses (*Figure 7F*), similar to

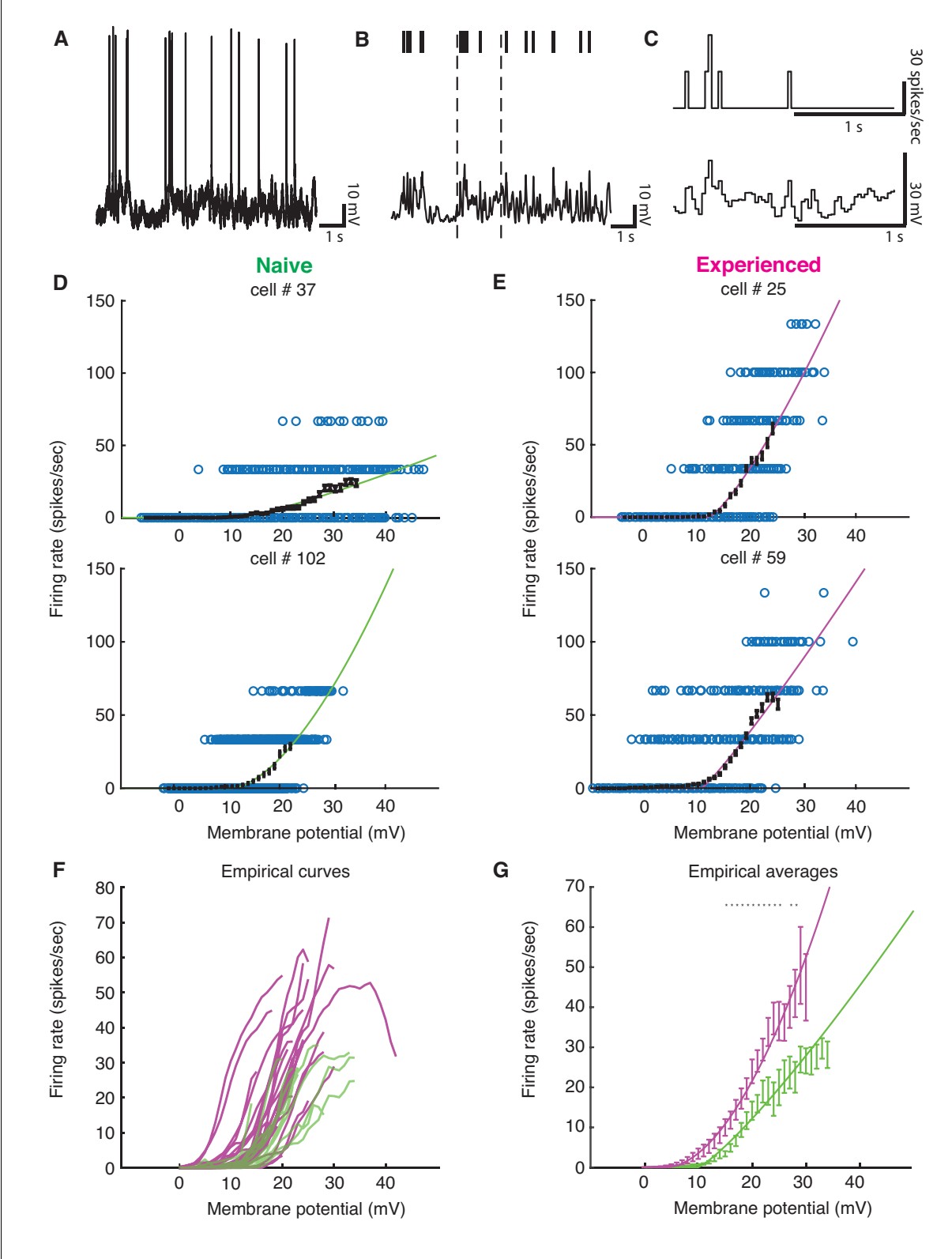

**Figure 6.** Developmental enhancement of input-output gain. (A) To calculate the voltage-to-firing rate transform, we (B) extracted spike times and used median filtering to remove spike waveforms from the membrane potential, and (C) averaged firing rate and spike-filtered membrane potential into 30 ms bins (*Priebe et al., 2004*; *Priebe and Ferster, 2005*; *Priebe and Ferster, 2006*). (D) Membrane potential and firing rate observations from two cells from naive and (E) two cells from experienced animals. Empirical means and standard error of the mean in sliding 1 mV windows (black) and a

*Figure 6 continued on next page*

*Figure 6 continued*

power law fit (*Priebe et al., 2004*) (naive are green, experienced are magenta) are shown for each cell. Empirical means are calculated up until first bin that does not contain 15 Vm, firing rate ordered pairs. (F) Empirical mean voltage-to-firing rate relationships for all cells (N = 16 naive, N = 29 experienced). (G) Mean and standard error of the mean empirical voltage-to-firing rate transforms for naive and experienced animals, averaging over cells; * indicates locations where a t-test shows a p-value of less than 0.05. Voltage values that had data for at least two cells are shown. Experienced voltage-to-firing rate transforms exhibited increased firing rates at a given voltage over a large membrane voltage range. Solid lines are power law fits, weighed by number of data points at each voltage.

The online version of this article includes the following figure supplement(s) for figure 6:

**Figure supplement 1.** Voltage-to-firing rate transforms for all cells in the study.

the actual data (*Figure 2EF* and *Figure 7CE*). Similarly, when these quantities were both selected from experienced cells, the simulations exhibited high direction selectivity and higher firing rates as in the actual data (*Figure 2EF* and *Figure 7CE*). Applying the experienced voltage waveforms to naive voltage-to-firing rate transforms resulted in a big increase in direction selectivity, even larger than those measured in mature animals (*Figure 7CD*), but the simulated firing rates were below those measured in the actual experienced animals (*Figure 7EF*). Applying the naive voltage wave-forms to experienced voltage-to-firing rate transforms did not produce substantial increases in direction selectivity compared to the naive case, although firing rates were elevated. Therefore, changes in the underlying selectivity of the voltage waveforms from naive to experienced was sufficient to confer increased direction selectivity values across these conditions, but the increased gain of the experienced voltage-to-firing rate transforms were needed to also confer the higher firing rates observed in experienced animals (*Figure 7CDEF*).

## Discussion

We studied the evolution of spatiotemporal receptive fields, excitability, and voltage-to-firing rate transformations over a short period of rapid visual development of direction selectivity in ferret visual cortex. Shape analysis of linear spatiotemporal receptive fields of simple cells showed that the elliptical subunits in experienced animals exhibited more eccentric profiles and contained energy at shorter latencies not seen in naive animals. This reorganization led to a significant increase in direction selectivity at the level of membrane voltage. Neurons in experienced animals exhibited hallmarks of increased cell-intrinsic excitability, such as a lower spike threshold. When analyzed with system-response methods, we found that these changes in excitability contributed to a substantial increase in the voltage-to-firing rate transform with experience. The full extent of increases in spiking activity could only be explained by considering both the enhancement in Vm responses and the enhancement of input-output gain.

### Extension and refinement of spatiotemporal receptive field structure

In the developing visual system, molecular cues and spontaneous activity (*Katz and Shatz, 1996*) shape the formation of initial circuitry and the emergence of initial cortical receptive field properties, including retinotopic organization (*Cang et al., 2008*) and orientation selectivity (*Chapman and Stryker, 1993*; *Crair et al., 1998*; *Li et al., 2006*; *Smith et al., 2018*) (although not its alignment across the two eyes, which requires experience *Wang et al., 2010*; *Gu and Cang, 2016*; *Whitney et al., 2019*). In cortex of ferrets (*Li et al., 2006*) and primates (*Hatta et al., 1998*) (but not mice: *Rochefort et al., 2011*), selectivity for direction of motion is not present at eye opening – neurons respond about equally to stimulation in two opposite directions – but develops over a period of about 2 weeks in ferrets and about 4 weeks in macaques.

The question of how a neural system can be responsive to a wide stimulus set and then become more restricted with experience reminds one of the 'overproduction and pruning' hypothesis (*Brown et al., 1976*; *Huttenlocher, 1979*; *Purves and Lichtman, 1980*; *Petanjek et al., 2011*), also called the 'stabilization' hypothesis (*Changeux and Danchin, 1976*). One might imagine that the connections that support multiple responses are present initially, and then, with activity-dependent development, 'inappropriate' connections are pruned away and reinforced connections are strengthened (*Meliza and Dan, 2006*). Alternatively, an early compact receptive field could become extended with visual experience, through addition of new inputs. Here, we found evidence for a

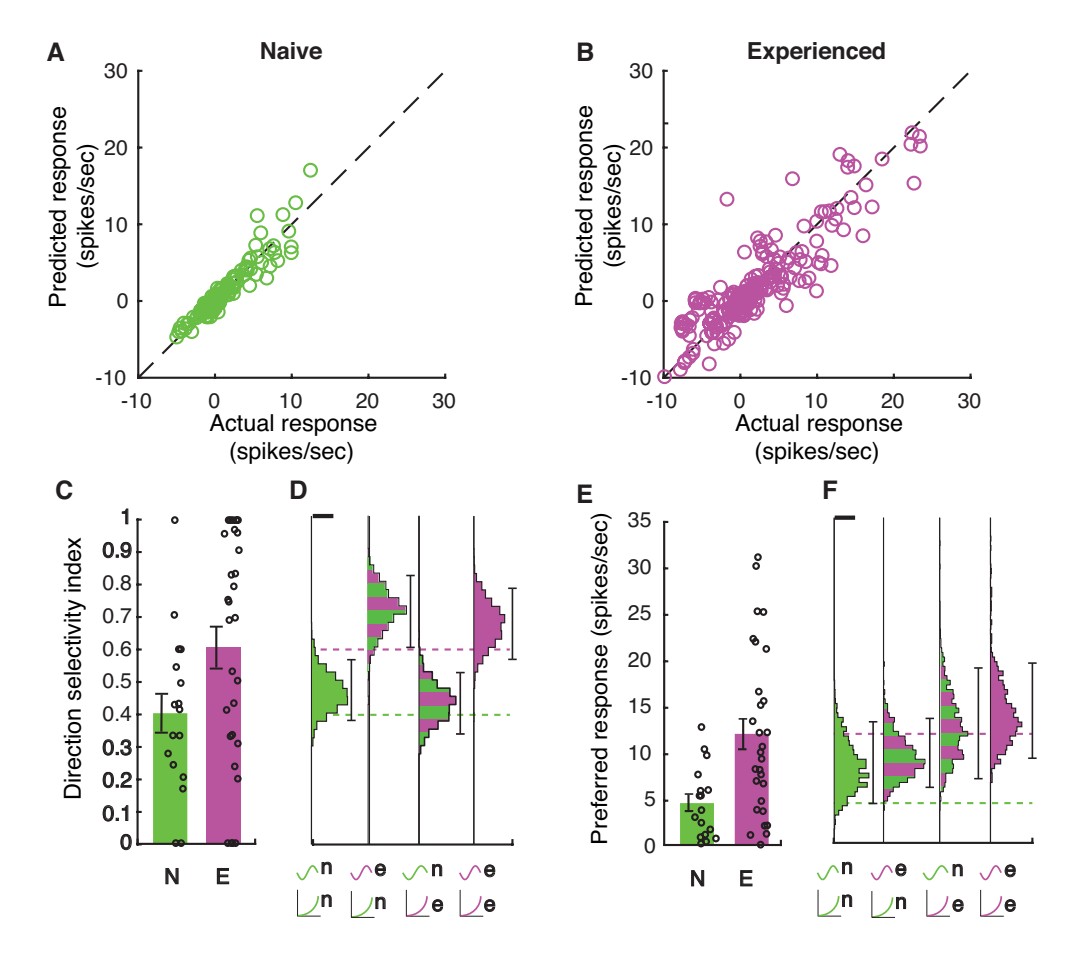

**Figure 7.** Increases in membrane potential selectivity and input-output gain are both required for mature direction selectivity and firing rates. (**A**) Actual mean spiking response rates for each stimulus plotted against the mean spiking response that is obtained by applying the voltage-to-firing rate transform to each stimulus's membrane potential record for all naive cells (N = 16), showing that the phenomenological model captures the voltage-to-membrane transfer well. (**B**) Same, for experienced cells (N = 29). Model captures the actual spiking responses well. (**C**) Actual direction index values derived from the mean responses (F0) of naive and experienced cells. (**D**) Simulations of what we would expect if naive or experienced waveforms were paired with naive or experienced voltage-to-firing rate transforms. Plots show vertical histograms (with 5–95 percentile error bars) of mean direction selectivity index values for 10,000 simulations of 20 cells for several cases: 1) randomly selected naive voltage waveforms applied to randomly selected naive voltage-to-firing rate transforms; 2) randomly selected experienced voltage waveforms applied to randomly selected naive voltage-to-firing rate transforms; 3) randomly selected naive voltage waveforms applied to randomly selected experienced voltage-to-firing rate transforms, and 4) randomly selected experienced voltage waveforms applied to randomly selected experienced voltage-to-firing rate transforms. Scale bar on top of histogram shows 100 simulations. Axis labels identify the model by the waveforms used (top symbol) and voltage-to-firing rate transform used (bottom symbol). Actual data values are shown as dashed lines. To achieve increases in direction selectivity, it was only necessary for the model to adopt the voltage response waveforms of the experienced animals: the increased gain of the experienced animals was not necessary. (**E+F**) Same as **C,D**, but response to preferred direction is shown. Scale bar shows 50 simulations. To achieve increases in firing rates observed in experienced animals, experienced voltage-to-firing rate transforms were necessary. In D,F, models can be compared to whether they match the individual cell data by seeing if the individual cell data mean across cells falls within the 5–95%-tile range (the distribution of means is sampled empirically through simulation). Overall, to achieve both increases in direction selectivity and firing rate that are observed in vivo, changes in voltage responses and voltage-to-firing rate transform were necessary.

reorganization process that combines some elements of both receptive field constriction and expansion. On the one hand, the mature receptive fields became elongated along the temporal axis, particularly by exhibiting responses with shorter response latencies that did not exist in the naive state. However, the mature receptive fields also became significantly narrow in space-time, providing evidence for a narrowing process. This process is reminiscent of the sharpening (narrowing) of spatial

receptive fields over the first weeks and months after the onset of experience (*DeAngelis et al., 1993a*; *Tavazoie and Reid, 2000*; *Kiorpes, 2015*; *Kiorpes, 2016*).

## Increased intrinsic excitability

As neurons mature, they must regulate their ion channels and synaptic connections in order to bring their spiking response rates into an appropriate physiological range (*Turrigiano and Nelson, 2004*). At eye opening, neurons in vivo exhibit weak spiking activity (*Clemens et al., 2012*) (this study) and both rats (*Kasper et al., 1994*) and ferrets (this study) exhibit wide action potentials. Further, as neurons are maturing during this stage just after eye opening, their input resistance also rapidly decreases (*Kasper et al., 1994*; *Desai et al., 2002*; *Wang et al., 2012*), which means the same input current will produce a smaller resulting voltage. Previous studies in rats have shown that layer 4 neurons add synapses during this time (*Desai et al., 2002*) and that spike threshold decreases (*Kasper et al., 1994*), presumably to bring the visual response rates up to mature values and to overcome the decreasing input resistance. In other feed-forward systems, the weight of feed-forward connections increases substantially: ~50 fold over the first week of vision at the retinogeniculate synapse (*Chen and Regehr, 2000*) and 2–4 fold from P4 to P15 in the barrel cortex (*Li et al., 2013*), presumably to allow mature selectivity and firing rates. Because subthreshold voltage responses increase over the first week of vision and because input resistance is going down, feed-forward synaptic contributions to visual cortex are also likely to be increasing, although we are not aware of direct measurements that would confirm this hypothesis. Increases in sodium channel density at the axon initial segment may underlie the enhancement in the maximum slope of Vm during an action potential (max dV/dt) and decreased threshold observed here (*Colbert and Pan, 2002*; *Kole et al., 2008*; *Hu et al., 2009*; *Grubb and Burrone, 2010*; *Kuba et al., 2010*).

While we observed an increase in input-output gain (in vivo), a reduced spike threshold (ex vivo), and evidence of a more excitable membrane (increased maximum dV/dt), we did not observe an overall increase in spiking in ex vivo current-firing rate (F-I) curves (*Figure 5L*). An F-I curve plots the cellular firing rate as a function of the amplitude of somatically injected current. The current-to-spiking transformation takes place in two distinct steps: in the first step the injected current elicits a change in Vm, a process gated by the cell's input resistance. Due to developmental decreases in input resistance we observed, the same current amplitude would elicit a smaller Vm change in the cells from experienced as compared to the naive animals. In the second step, a given level of Vm elicits the firing of a certain number of spikes, a process gated by the mechanisms of spike generation such as threshold voltage. Our in vivo data show that this Vm-to-spike transformation is steeper in the experienced animal, possibly mediated by lowered spike threshold. During the measuring of F-I curves, these two factors possibly cancel each other, resulting in F-I curves indistinguishable across the two developmental stages. In addition, stronger visually evoked synchronous synaptic currents in the experienced state might help overcome the effect of lower input resistance, resulting in higher Vm responses after all. For these reasons, the lack of enhancement in F-I curves obtained from slices does not contradict the strong enhancement in intrinsic excitability, as evidenced by the steeper Vm-to-spike curves and lower thresholds.

## Developmental changes in input-output gain

The spike threshold non-linearity contributes substantially to cortical direction selectivity (*Reid et al., 1987*; *Jagadeesh et al., 1993*; *Priebe et al., 2004*; *Priebe and Ferster, 2005*). Before these experiments, it was possible to imagine that much of the developmental increase in cortical direction selectivity in ferret might be due not to changes in subthreshold inputs, but rather purely through changes in effective threshold or the voltage-to-firing rate transform (*Van Hooser et al., 2014*; *Christie et al., 2017*). The experiments here show that increased input-output gain does contribute to the increased firing rates that are observed in experienced animals, but that increased input-output gain did not, by itself, account for the increased direction selectivity index values (see *Figure 7CD*). Instead, changes in the underlying voltage responses were necessary to confer increased direction selectivity.

## Possible synaptic bases of spatiotemporal receptive field reorganization

It is interesting to consider the possible synaptic bases for the expanded receptive fields we observed here, as we cannot directly examine the physiological properties of the synapses that underlie these receptive fields at this time. A variety of mechanisms seem plausible given the data.

First, the receptive field expansion that we observed might be due to recruitment of additional synapses between LGN neurons and cortical neurons, such that newly connected neurons are providing inputs at the latencies and positions that are newly present in the spatiotemporal receptive fields of mature neurons. These results fit nicely with prior anatomical observations that the number of excitatory synapses being formed in ferret and human visual cortex during this time are greatly increasing (*Huttenlocher, 1990*; *Durack and Katz, 1996*; *Ruthazer and Stryker, 1996*; *White et al., 2001*; *Erisir and Harris, 2003*; *White and Fitzpatrick, 2007*), and are consistent with the idea that precise growth may be an important principle of the development of some neural circuits.

A second hypothesis is that the maturation of receptive field properties of already-connected LGN neurons or cortical neurons might cause the spatiotemporal receptive field to expand, and confer direction selectivity even without such selectivity in the antecedent inputs. For example, LGN neuron latency drops during the first weeks of visual experience *Tavazoie and Reid, 2000*; therefore, it is possible that changes in receptive field properties of already-connected LGN cells that exhibit purely center surround receptive fields could fill out the earlier latency regions of the experienced spatiotemporal receptive fields. Experiments that measure thalmocortical inputs to cortical neurons (*Alonso and Reid, 1995*; *Swadlow et al., 2005*; *Lien and Scanziani, 2018*), performed in developing animals, could support or reject this hypothesis. Regardless of whether LGN inputs are altered or constant, there is clear evidence linking the spatiotemporal structure of LGN inputs to direction selectivity in V1 in mice and carnivores (*Kremkow et al., 2016*; *Lien and Scanziani, 2018*).

A third hypothesis is that a change in the relative contributions of excitation and inhibition within the cortex underlie the maturation we observed. Although the data here force us to reject the hypothesis that the change in direction selectivity is purely due to postsynaptic changes in excitability, it remains possible that there might be no changes in LGN inputs and that amplified firing from recurrent connections underlie the increases in Vm selectivity (*Douglas et al., 1995*; *Suarez et al., 1995*; *Christie et al., 2017*). Under this view, contributions could increase from the excitatory side such that the enhanced gain in cortical neurons would recurrently drive changes in the synaptic inputs, leading to additional direction selectivity. Alternatively, removal of cortical shunting inhibition or alterations in the timing of inhibition (*Nelson et al., 1994*; *Priebe and Ferster, 2005*; *Wilson et al., 2018*), which impact direction selectivity, could also sculpt these developing receptive fields.

Further studies will be needed to tease apart the contributions of feed-forward and recurrent connections to direction selectivity.

# Materials and methods

**Key resources table**

| Reagent type (species) or resource | Designation | Source or reference | Identifiers | Additional information |
|---|---|---|---|---|
| Software, algorithm | Matlab | The MathWorks, Natick, MA | RRID:SCR_001622 | |
| Software, algorithm | GitHub | GitHub | RRID:SCR_002630 | |
| Software, algorithm | Psychophysics Toolbox | Psychtoolbox.org | RRID:SCR_002881 | |
| Software, algorithm | Spike2 | Cambridge Electronic Design | | |
| Other | Spyder Express 3 | Datacolor | | |

*Continued on next page*

*Continued*

| Reagent type (species) or resource | Designation | Source or reference | Identifiers | Additional information |
|---|---|---|---|---|
| Strain, strain background (*Mustela putorius furo*) | Ferrets | Marshall Bio-resources | 'Conventional' colony | Females |
| Other | Electrode Puller | Sutter Instrument Company | P-97 with box filament | |
| Other | Micromanipulator | Sutter Instrument Company | MP-285 | |
| Other | Microelectrode amplifier | Axon Instruments (now Molecular Devices) | AxoClamp-2B | |
| Other | Multifunction data acquisition system | Cambridge Electronic Design | Micro1401 | |

## General design

All experimental procedures were approved by the Brandeis University Animal Care and Use Committee and performed in compliance with National Institutes of Health guidelines. In vivo intracellular recordings were carried out in two groups of ferrets (*Mustela putorius furo*): 'naive' (postnatal day (P) 30–34; n = 10) and 'experienced' (P40-60; n = 11).

## Sex as a variable

Only female ferrets were used because housing mature male and female ferrets in the same room causes significant hormonal stress for the female animals if they are not allowed to mate.

## Surgical preparation

Ferrets were sedated with ketamine (20 mg/kg intramuscular) and given atropine (0.16–0.8 mg/kg intramuscular) and dexamethasone (0.5 mg/kg intramuscular) to reduce bronchial and salivary secretion and to reduce inflammation. Ferrets were next anesthetized with a mixture of isoflurane, oxygen, and nitrous oxide through a mask while a tracheostomy was performed. Ferrets were then ventilated with 1.5–3% isoflurane in a 2:1 mixture of nitrous oxide and oxygen. A cannula was inserted into the intraperitoneal (IP) cavity for delivery of neuromuscular blockers and Ringer's solution (3 ml/kg/hr). The animals were next inserted in a custom stereotaxic frame that did not obstruct vision. All wound margins were infused with bupivacaine. Silicone oil was placed on the eyes to prevent corneal damage. An incision was made on the scalp, a craniotomy performed, and a durotomy performed with a sharp needle (31 gauge). Before recording commenced, ferrets were paralyzed with the neuromuscular blocker gallamine triethiodide (10–30 mg/kg per hour) through the IP cannula to suppress spontaneous eye movements, and the nitrous oxide-oxygen mixture was adjusted to 1:1. The animals' ECG was continuously monitored to ensure adequate anesthesia, and the percentage of isoflurane was increased if the ECG indicated any distress. Body temperature was maintained at 37°C.

## In vivo intracellular electrophysiology

Intracellular recordings were performed using sharp microelectrodes (borosilicate glass, BF100-50-10; Sutter Instruments, CA) with a resistance of 80-120MO when filled with 3M KCl and 5% Neurobiotin at the tip. The electrodes were lowered into the primary visual cortex using a Sutter P-285 micromanipulator. The intracellular potentials were recorded using an AxoClamp 2B amplifier (Molecular Devices, CA) in the bridge mode, and subsequently low-pass filtered at 3 kHz, digitized at 8–11 kHz (micro1401, Cambridge Electronics Design), and stored on a personal computer (PC) hard drive. V1 neurons were recorded between the depths of 230 and 1100 µM, and cells were later classified into simple and complex types based on spike modulation ratio (*Figure 2—figure supplement 1*).

## Visual stimuli

Visual stimuli were created in MATLAB with the Psychophysics Toolbox (*Brainard, 1997*; *Pelli, 1997*; *Kleiner et al., 2007*) on a Macintosh Pro running OSX and displayed on a Dell monitor 1704FPVt (40 cm viewing distance). Gratings were shown in pseudorandom order at temporal frequencies that varied between 2–8 Hz and a spatial frequency of 0.08 cycles per degree, which is in the middle of the maximal response functions for animals in this age range (*Li et al., 2006*; *Popović et al., 2018*). Sparse noise stimuli for reverse correlation analysis were 1-dimensional bar stimuli, rotated so that the bar orientation matched the preferred orientation of each cell (*Priebe and Ferster, 2005*; *Rust et al., 2005*). Bar width varied from 1.55 to 3.1 degrees of visual angle, and bar luminance values were updated each 100 ms to be black (10% probability), white (10% probability), or gray (80% probability).

## Acute slice electrophysiology

Ferrets (n = 9) were split into naive (n = 6, age P28-32, 29 cells) and experienced (n = 3, age P41-44, 17 cells) groups. Animals were injected with an intraperitoneal dose of Beuthanasia-D (1 μL/g; 390 mg/mL sodium pentobarbital, 50 mg/mL sodium phenytoin) to induce acute respiratory arrest. Transcardial perfusion was immediately performed using 0.22 M sucrose artificial cerebrospinal fluid (aCSF) that had been chilled to 4°C and aerated in 95% carbon dioxide/5% oxygen gas mixture for 15 min. The full composition of the solution used for perfusion and slicing was 220 mM sucrose, 26 mM NaHCO$_3$, 3 mM KCl, 5 mM MgCl, 1.25 mM NaH$_2$PO$_4$, 1.0 CaCl$_2$, and 10 mM dextrose. Following decapitation using a small animal guillotine, the brain was extracted, separated into hemispheres and cut again into caudal coronal blocks to excise the occipital lobe before slicing into 330 μm coronal sections on a Leica VT-1000s vibratome at high vibrating frequency in sucrose cutting solution. Sections were retained if fully intact and obtained in the second through the fifth passes in the slicing sequence (i.e. not more than 1650 microns from the extremity of the occipital pole) and most recorded neurons were taken from the second and third slices. Each slice was immediately incubated in aerated standard aCSF (124 mM NaCl, 2.5 mM KCl, 2 mM MgSO$_4$, 1.25 mM NaHPO$_4$, 2 mM CaCl$_2$, 26 mM NaHCO$_3$, 10 mM dextrose, ph 7.4, mOsm 318 ± 2) at 34–35°C for 15–30 min and then allowed to recover at room temperature in the same solution for 1 to 1.5 hr. Slices were maintained in an aerated incubation chamber at room temperature until needed for recording when they were transferred to the perfusion chamber and maintained at a temperature of 35° to 37°C during recording.

Current clamp measurements of single neuron intrinsic excitability were obtained in standard ACSF containing 20 μM 6,7-dinitroquinoxaline-2,3-dione (DNQX), either 25 μM of D-2-amino-5-phosphonopentanoate (D-AP-V) or 50 μM of racemic mixture AP-V, and 20 μM of picrotoxin, synaptic blockers of AMPA, NMDA, and GABA, respectively. A small ad-hoc DC current was injected for all cells to set baseline membrane potential to −60 mV. A series of current pulses were tested over 10 to 15 evenly spaced step sizes between 0 and 400 pA. Single current pulses were 300 too 400 ms in duration over a 1 s acquisition frame with stimulation onset delayed by 50 ms. A 3 s minimum delay was used between all single pulse acquisition frames. The order of current step amplitudes was pseudorandomly interleaved and a minimum of three repetitions were acquired for each current level.

## Data analysis

For Vm analysis, spikes were removed from the traces using an 8 ms median filter (*Ferster and Jagadeesh, 1992*). Responses to each stimulus were calculated relative to a baseline that was calculated in the 5 s preceding each stimulus (or shorter for a few cases where the interstimulus interval was shorter than 5 s). The baseline was defined as the 20th percentile of the data points in this time interval, to discount any spontaneous activity that was more prevalent especially in the older animals; manual inspection showed that this value was very close to what the experimenter would have selected on a case-by-case basis. For spike analyses, spikes were detected using a manually set threshold on the Vm trace for each recording epoch. For each cell, we examined both the mean response to drifting grating stimulation (F0) as well as the modulation at the stimulus frequency (F1). If a cell's F1 response was greater than the mean response (F0), then the cell was declared to be a 'simple' cell. Direction selectivity was examined in recorded cells that exhibited significant variation

across all stimuli by an ANOVA test, and tuning curves were fit with a double Gaussian (*Mazurek et al., 2014*). The direction selectivity index (DSI) was defined to be (Rp(θ) - Rn(θ))/Rp(θ), where Rp(θ) is the fitted response in the preferred direction (θ) and Rn is the response in the opposite direction 180˚ + θ.

## Reverse correlation with sparse noise

Vm traces median-filtered at 8 ms to remove spikes as above. For each spatial location, the stimulus trace was cross-correlated (*Citron et al., 1981*) with the median-filtered Vm trace using the xcorr function in Matlab. Correlation values were plotted as a function of lag time with respect to stimulus onset (time 0). Attempts to sharpen (or whiten) the reverse correlation to remove the blurring from the step-function-like stimuli typically resulted in unstable filters, so only the raw stimulus-voltage correlation waveform is plotted here, as these measurements were always stable.

### Subunit analysis

Once the Vm STRFs were computed, pixel clusters representing each ON and OFF subunit were detected through connected component analysis using the *bwconncomp* function in MATLAB. Only the brightest and the darkest 5% of pixels from the STRF images were analyzed using *bwconncomp* for ON and OFF subunits, respectively. To eliminate random pixel clusters spuriously detected by the method, only pixel clusters containing at least 20 pixels were accepted as subunits, and an ellipse was drawn through each accepted pixel cluster to represent a subunit. Next, using the *regionprops* function in MATLAB, the following parameters were extracted for each ellipse: orientation, major axis length, minor axis length, eccentricity and area. Further, using these parameters and the geometric properties of an ellipse, the following parameters were also calculated: spatial extent (the projection of the ellipse on the x or spatial axis), temporal extent (the projection of the ellipse on the y or time axis), maximum latency (the upper bound of the ellipse on the time axis) and minimum latency (the lower bound of the ellipse on the time axis). Values for each of the above nine parameters from all ON and OFF subunits from a single cell were averaged to obtain cell-average values.

Because visual stimulation evoked stronger Vm responses in the experienced group, the weaker responses in the naive group might have yielded less reliable correlations to the stimulus contrast, leading to spurious differences in STRF shape parameters. To test that possibility, we compared the absolute peak correlation values of the ON (positive) and OFF (negative) subunits across the two developmental groups, and found that on average the peak correlation values were indistinguishable between the two groups (Naive_ON x Naive_OFF x Exp_ON x Exp_OFF; one-way ANOVA, p>0.05). This implies that the peaks and troughs of the ON and OFF subunits were separated from the correlation floor equally well in the naive and the experienced groups, thereby making it unlikely that weaker correlations might have led to unreliable STRFs in the naive group.

Two-dimensional Fourier transforms of the Vm STRFs were computed using the *fft2* function in MATLAB. The resulting STRFs in the frequency-domain, represented with spatial frequencies along the x-axis and temporal frequencies along the y-axis, had four quadrants around the origin representing 0 spatial and temporal frequencies (*Priebe and Ferster, 2005*). The quadrants diagonally opposite to each other represented the same pattern of amplitudes. The quadrant with positive spatial and temporal frequencies represented one direction of motion, while the quadrant with positive spatial and negative temporal frequencies represented the opposite direction of motion. All FFT amplitudes (within spatial frequency range 0 to 0.2 cycles/degrees, and temporal frequency range 0 to −10 or 10 Hz) in these two quadrants were summed to obtain a predicted response to the forward and reverse direction of motion, respectively (*Priebe and Ferster, 2005*). The larger of the two responses was taken as the preferred response ($R_{pref}$) and the smaller was taken as the null response ($R_{null}$). Predicted Vm DSI was then calculated as $DSI_{pr}$ = ($R_{pref}$ - $R_{null}$)/$R_{pref}$.

## V-F transform

Vm signals were filtered with a 60 Hz Savitzky-Golay filter to remove a small 60 Hz component, and spikes were filtered out of Vm by applying an 8 ms median filter (*Finn et al., 2007*) and baseline subtraction as described above. Next, a set of ordered pairs of Vm values and corresponding firing rate values were created by calculating the mean Vm and mean firing rate in non-overlapping 30 ms

windows that were run across the data (*Priebe and Ferster, 2005*; *Priebe and Ferster, 2006*). The relationship between these pairs (Vm and firing rate) for each cell was fit by a power law function: FR (V)=b * *rectify*(V-Vth)$^{\alpha}$, where $\alpha$ was constrained to be between 1 and 4 and rectify(x):=x, if x $\geq$ 0; 0, if x < 0. Empirical data was plotted against these fits by binning observations into 1 mV bins and computing the mean and standard error of the mean.

### Spike kink analysis
Spike 'kinks' (a particular definition for spike threshold) were calculated for spikes following the algorithm by *Azouz and Gray, 1999*. In brief, dV/dt was calculated by dividing the difference between successive samples by the time interval between them. The spike kink voltage was defined to be the point at which dV/dt exceeded 0.033 of the maximum value.

## Acknowledgements
We thank Zoey Keeley and Daniel Shin for help with figure layout, and members of the Van Hooser lab for comments and feedback during the project and on the manuscript.

## Additional information

### Funding

| Funder | Grant reference number | Author |
| --- | --- | --- |
| National Eye Institute | EY022122 | Arani Roy<br>Jason J Osik<br>Benyamin Meschede-Krasa<br>Wesley T Alford<br>Daniel P Leman<br>Stephen D Van Hooser |

The funders had no role in study design, data collection and interpretation, or the decision to submit the work for publication.

### Author contributions
Arani Roy, Conceptualization, Software, Formal analysis, Investigation, Methodology, Writing - original draft, Writing - review and editing; Jason J Osik, Software, Formal analysis, Investigation, Methodology; Benyamin Meschede-Krasa, Formal analysis, Investigation, Methodology; Wesley T Alford, Investigation, Methodology; Daniel P Leman, Formal analysis; Stephen D Van Hooser, Conceptualization, Data curation, Software, Formal analysis, Supervision, Funding acquisition, Writing - original draft, Project administration, Writing - review and editing

### Author ORCIDs
Stephen D Van Hooser (ID) https://orcid.org/0000-0002-1112-5832

### Ethics
Animal experimentation: This study was performed in strict accordance with the recommendations in the Guide for the Care and Use of Laboratory Animals of the National Institutes of Health. All of the animals were handled according to approved institutional animal care and use committee (IACUC) protocols of Brandeis University (19010, 16003, 13011). All procedures were performed under isoflurane anesthesia and every effort was made to minimize suffering.

### Decision letter and Author response
Decision letter https://doi.org/10.7554/eLife.58509.sa1
Author response https://doi.org/10.7554/eLife.58509.sa2

## Additional files

### Supplementary files
- Transparent reporting form

### Data availability

Data is available at our website at http://data.vhlab.org and at Dryad (https://doi.org/10.5061/dryad.n02v6wwvb). Code is available at http://code.vhlab.org (copy archived at https://github.com/elifesciences-publications/vhlab-publishedstudies).

The following dataset was generated:

| Author(s) | Year | Dataset title | Dataset URL | Database and Identifier |
|---|---|---|---|---|
| Roy A, Osik JJ, Meschede-Krasa B, Alford WT, Leman DP, Van Hooser SD | 2020 | Synaptic and intrinsic mechanisms underlying development of cortical direction selectivity | https://doi.org/10.5061/dryad.n02v6wwvb | Dryad Digital Repository, 10.5061/dryad.n02v6wwvb |

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
