## [Decision Letter]

**Acceptance summary:**

The authors have done a great job revising the paper. The new paper organization makes a much clearer distinction between experimental findings and interpretation. The Results section is now fully focused on describing the developmental changes that they observe while the Discussion section describes three possible alternative interpretations of the findings.

**Decision letter after peer review:**

Thank you for submitting your work entitled "Synaptic and intrinsic mechanisms underlying development of cortical direction selectivity" for consideration by *eLife*. Your article has been evaluated by three reviewers, one of whom is a member of our Board of Reviewing Editors, and the evaluation has been overseen Laura Colgin as the Senior Editor.

This manuscript has been read by three reviewers. All three agree that this is an excellent study with high quality data addressing he important questions regarding the mechanisms underlying the maturation of receptive fields in ferret visual cortex. However, there were significant concerns of the context in which these data were presented. These concerns can be addressed with significant revisions to text and some additional analysis.

Reviewer #1:

This study represents a careful dissection of the maturation of receptive fields in ferret primary visual cortex. It has been well-established that direction selective responses in ferret cortex emerge in an activity dependent manner after eye opening. The authors use in vivo sharp electrode recordings to gain insights into the changes in receptive field properties that underlie the maturation of directional tuning. Specifically, they first show that both subthreshold changes in membrane potential as well as spiking output increase their direction selectivity tuning. Second, they use random visual stimuli to characterize the space-time receptive fields (STRFs) and compare properties for both immature and mature simple cells. From these data they see there is both an expansion of compact receptive fields and an enhanced tilt in space-time. Third, they use in vivo and in vitro recordings to show that there is an increase in the excitability of cells by assessing action potential waveforms. Fourth they compute voltage-> firing rate transformation (i.e. input-output gain, a term first introduced by classic papers by Priebe and Ferster). Finally, they use a simple model to show that need both changes in receptive fields and input-output gain functions to explain the change in tuning.

The paper is clearly written (though the Abstract is a bit obtuse – see below) and the data clearly presented. The data is gorgeous and based on very difficult experiments. The authors are to be commended.

1) The Abstract can really only be understood by people within the field – particularly the sentences about the STRFs (e.g. "expansion of space-time receptive fields along the temporal dimensions" and "marked loss of inputs with certain space-time selectivities").

2) In Figure 5J-L, the authors argue that the change in spike waveform seems to balance out the decrease in input resistance and increase capacitance associated with development due to the enlargement of cells. So, it is not clear whether there is a change in excitability contributed to the enhanced input-output gain.

Reviewer #2:

In this paper, the authors studied mechanisms underlying the development of direction selectivity in ferret visual cortex. They performed intracellular recording to reveal both synaptic and intrinsic membrane properties that are correlated with direction selectivity development. Overall, the paper addressed an important question; the data were of high quality; and presentation was largely clear. However, the authors fell short in connecting the synaptic changes (spatial temporal receptive fields) with changes in direction selectivity, which can be potentially corrected by additional analysis.

1) The main finding of the paper is changes in the spatial temporal receptive fields (STRF; Figure 4), which was interpreted as changes in synaptic connections (Figure 1). Specially, the authors concluded that there was both pruning of existing connections (because of the narrowing of minor axis) and forming of new connections (increase in major axis and decrease of latency). However, as the authors mentioned, dLGN cells' receptive fields become smaller and faster, which could account for much of the changes in cortical receptive fields. Can the authors perform additional analysis using existing LGN data to determine how much of the observed changes can be explained by changes in the dLGN or more upstream, without a change in synaptic connection?

2) The most dramatic change to me is actually the STRF orientation (Figure 4C, D), the extent of which does not appear to be reflected by the quantification in Figure 4A. Please double check the analysis. More importantly, the authors did not provide any evidence connecting STRF structure with direction selectivity, despite how they framed the study (Figure 1). Did the STRF predict the preferred direction and DSI in adult animals? In young animals, this does not seem to be the case for DSI (Figure 4B), perhaps because most of them did not have any tilt.

3) Assuming STRF structure indeed determines direction selectivity (to be shown by the authors), what changes could mediate the appearance of the adult tilt? Synaptic reorganization, changes in dLGN response temporal profile, or conduction delay? Some discussion are needed given its importance.

Reviewer #3:

This is an excellent and important piece of work that investigates the development of direction selectivity in visual cortex. The mechanism underlying cortical direction selectivity has been a topic of great interest for several decades and attracted the attention of many different laboratories across the world. One of the most powerful approaches to study input-output neuronal transformations in the brain is to perform in vivo intracellular recordings. However, this approach is technically very difficult and, to my knowledge, it has only been used to study cortical direction selectivity in adult animals. Remarkably, the authors succeeded at performing these experiments in immature developing brains and the data that they collected is simply beautiful. Just because the data is unique and obtained with such a powerful approach, the study is of great value to the scientific community. In addition, the high quality of the data and the unexpected findings increase even further the scientific impact.

My comments should be seen more as suggestions to improve an already strong paper rather than criticisms. Most of my comments relate with the interpretation of the results and are relatively minor.

1) Abstract. The Abstract motivates the work by offering two alternative mechanisms of wiring development, elimination of redundant connections or formation of new connections. However, the general reader may think that this motivation is misleading since the development of neuronal connections frequently involves a combination of both. For example, each immature LGN neuron receives multiple retinal inputs that are eliminated as the brain matures. In addition, the remaining connections become stronger by making new connections (i.e. new synapses).

2) Abstract. The parallel that the Abstract makes between elimination/formation of connections and input/intrinsic changes does not work well. The mixed mechanism involving changes in input connectivity and intrinsic excitability is different from a mixed mechanism involving elimination and formation of connections. The authors may want to consider changing the motivation of the work (e.g. nobody knew the intracellular changes underlying the development of direction selectivity until the authors performed these experiments).

3) Abstract. It is unclear whether the authors can interpret the changes that they observe in the spatiotemporal receptive field as formation of new connections, as they claim in the Abstract. Changes in the temporal dimension of the spatiotemporal receptive field could be caused by combined changes in the receptive field size, time course and synaptic strength of already-connected inputs. A paragraph of the discussion states this wonderfully (quoted below) but the rest of the paper (including the Abstract) gives the impression that the results provide a direct demonstration of changes in the number of inputs during development. This could be misleading particularly for the non-specialized reader. Quote of paragraph: "Alternatively, maturation of receptive field properties of already-connected LGN neurons or cortical neurons might cause the spatiotemporal receptive field to expand, and confer direction selectivity even without such selectivity in the antecedent inputs. […] Or, perhaps connections from LGN cells with reduced latencies do not form until after the onset of experience".

4) Introduction and Results. Several sections of the paper appear to imply that the results provide direct evidence for formation and elimination of synaptic inputs. However, as the authors acknowledge in the Discussion, this is one of several possible interpretations. Therefore, it is probably a good idea to make it clear early in the paper that the elimination/formation of inputs is not a direct conclusion from the results but a possible interpretation. For example, the authors state: "…became more elongated along the temporal axis, corroborating the idea of recruitment of new synaptic inputs. However, the subunits also became significantly narrower along their preferred space-time axis, consistent with pruning of synaptic inputs with certain spatiotemporal properties". The term "consistent with" is probably more appropriate than "corroborating". "Corroborating" may lead the non-specialized reader to believe that the authors are providing a direct demonstration of input recruitment.

5) Changes in the relative strength of intracortical inhibition could be proposed as another possible interpretation of the results and should probably be discussed. Several studies have demonstrated that intracortical inhibition plays an important role in cortical direction selectivity including those from Sacha Nelson (the PhD advisor of the corresponding author) and, more recently, David Fitzpatrick (the postdoctoral advisor). An alternative interpretation of the results is that the receptive field maturation that they observe originates from changes in the relative strength of excitation and inhibition. For example, the much larger increase in spiking than Vm response during development could reflect a release from shunting inhibition as cortical neurons mature. This mechanism could also explain why the spiking response in the null direction is weaker in naive than mature ferrets. The developmental increase in response to the null direction may indicate that stimulus-driven excitation becomes stronger than stimulus-driven inhibition during development. Inhibition could also be relatively stronger in immature brains because cortical neurons spike less but each spike from inhibitory neurons has a relatively bigger impact in the cortical response because excitation is weak.

6) It may be a good idea to report the signal to noise ratio of the receptive field measurements. This is important to fully disregard the possibility that the reported differences are not partly due to differences in signal to noise between naïve and experienced animals. The signal to noise should be lower in naïve animals because stimuli drive weaker fluctuations in Vm and spike rate than in experienced animals. This is not a concern in the interpretation of the results because the authors did not find differences in area, spatial extent and temporal extent of the receptive fields between naïve and experienced animals. However, reporting signal to noise measurements somewhere in the Materials and methods section is important to fully eliminate this concern. If there is a difference in signal to noise, the authors can control for a possible contribution of this difference in their comparison by subsampling spikes in experienced animals at the same rate as in naïve animals and show that the most important results (e.g. changes in eccentricity) hold.

7) Results. The following statement is another example of a possible misleading conclusion: "…In sum, following visual experience, the Vm receptive fields exhibited marked reorganization… and recruitment of lower latency inputs". "recruitment of lower latency inputs" seems to imply that the results directly show this. However, recruitment of lower latency inputs is one of several possible interpretations. One interpretation is that already-connected LGN inputs become stronger and faster in response latency. It would be very interesting to see the relation between response strength and latency across the entire dataset. One possibility is that response strength and latency are negatively correlated and both change together during development.

8) It is very impressive to see all the pronounced and clear changes that the authors report in just 14 days of development. Great work!

Some references that should probably be cited:

Lien and Scanziani, 2018. To my knowledge, this is the only study measuring directly the thalamic inputs to directional selective cortical cells. It shows that, at least for some cortical cells, direction selectivity can be explained simply by the addition of thalamic inputs with different response time courses.

Kremkow et al., 2016. This study shows that rapid changes in direction preference within 100-200 microns of horizontal cortical distance are closely associated with rapid changes in receptive field position. This finding is also consistent with the notion that changes in direction preference are associated with changes in the receptive field position of thalamic inputs.

Wilson, Scholl and Fitzpatrick, 2018. There are many papers showing the importance of intracortical inhibition in generating cortical direction selectivity. Wilson et al., 2018, should probably be cited because it is the most recent one on this topic and uses ferrets, the animal model chosen by the authors.

---

## [Author Response]

Reviewer #1:[…] The paper is clearly written (though the Abstract is a bit obtuse – see below) and the data clearly presented. The data is gorgeous and based on very difficult experiments. The authors are to be commended.1) The Abstract can really only be understood by people within the field – particularly the sentences about the STRFs (e.g. "expansion of space-time receptive fields along the temporal dimensions" and "marked loss of inputs with certain space-time selectivities").

We have now completely re-written the Abstract to address the concerns of all 3 reviewers, and have removed the overtly technical terms as far as practicable.

2) In Figure 5J-L, the authors argue that the change in spike waveform seems to balance out the decrease in input resistance and increase capacitance associated with development due to the enlargement of cells. So, it is not clear whether there is a change in excitability contributed to the enhanced input-output gain.

We have now removed the interpretive sentences from the Results section pertaining to this figure, because it was causing some confusion. Instead we have now added a more detailed elaboration on this point in the Discussion section, under the subsection “Increased intrinsic excitability”. We quote the relevant paragraph here:

“While we observed an increase in input-output gain (in vivo), a reduced spike threshold (ex vivo), and evidence of a more excitable membrane (increased maximum dV/dt), we did not observe an overall increase in spiking in ex vivo current-firing rate (F-I) curves (Figure 5L). […] For these reasons, the lack of enhancement in F-I curves obtained from slices does not contradict the strong enhancement in intrinsic excitability, as evidenced by the steeper Vm-to-spike curves and lower thresholds.”

Reviewer #2:[…]1) The main finding of the paper is changes in the spatial temporal receptive fields (STRF; Figure 4), which was interpreted as changes in synaptic connections (Figure 1). Specially, the authors concluded that there was both pruning of existing connections (because of the narrowing of minor axis) and forming of new connections (increase in major axis and decrease of latency). However, as the authors mentioned, dLGN cells' receptive fields become smaller and faster, which could account for much of the changes in cortical receptive fields. Can the authors perform additional analysis using existing LGN data to determine how much of the observed changes can be explained by changes in the dLGN or more upstream, without a change in synaptic connection?

We agree that this is a critical next question, and this is the subject of an ongoing study in the lab that is recording from populations of LGN cells while ferret cortex acquires direction selectivity. We hope this study will be ready for publication in 6-12 months but it is a separate and complex effort (LGN cells in the same animal are studied before and after visual experience). We agree that this is a really interesting question to tackle but it is a separate study that is similar in size to the present work, and following a population over time in the same animal adds a degree of challenge that takes time.

2) The most dramatic change to me is actually the STRF orientation (Figure 4C, D), the extent of which does not appear to be reflected by the quantification in Figure 4A. Please double check the analysis. More importantly, the authors did not provide any evidence connecting STRF structure with direction selectivity, despite how they framed the study (Figure 1). Did the STRF predict the preferred direction and DSI in adult animals? In young animals, this does not seem to be the case for DSI (Figure 4B), perhaps because most of them did not have any tilt.

We agree that it is important to connect the STRF parameters and direction selectivity. First, in our initial submission, we attempted to address this in Figure 4B and in the subsection “Reorganization of simple cell spatiotemporal receptive fields following visual experience”. These show and describe correlations between all of the STRF parameters and direction selectivity measurements in naïve and experienced cells. Our analysis indeed finds that STRF subunit shape parameters such as eccentricity and latency correlate with the cells’ actually measured DSI values in ways that corroborate our idea that higher eccentricity and faster responses correspond with stronger direction selectivity. Second, to make this relationship even clearer, we have now added another analysis as a new Figure 4—figure supplement 1. In this analysis, we take advantage of Fast Fourier Transform to convert the STRFs into the frequency domain, and generate predictions of DSI from the resulting images (following the Methods in Priebe and Ferster, 2005). This analysis also shows that the Vm DSI values predicted from the STRFs are linearly correlated to the actually measured Vm DSI values (for the linear F1 component), with an R^2^ value of 0.32. This suggests that about one third of the variance in direction selectivity of simple cells can be explained by variance in linear spatiotemporal summation properties as captured in the STRFs. Taken together, we believe these analyses provide incontrovertible evidence that the STRF structure has a strong influence on the cells’ direction selectivity.

The reviewer has also mentioned that the orientation/tilt of the STRF subunits should be the most dramatic effect. We have checked our analysis and the results reported in Figure 4A are accurate. However, there is an outlier that causes the “orientation-DSI” relationship in Figure 4B to be not significant. We give the results with and without the outlier (without the outlier, r^2^ = 0.26 and p≤0.03).

3) Assuming STRF structure indeed determines direction selectivity (to be shown by the authors), what changes could mediate the appearance of the adult tilt? Synaptic reorganization, changes in dLGN response temporal profile, or conduction delay? Some discussion are needed given its importance.

Please see our response to comment 2 above. Furthermore, we have now added substantial new texts to the Discussion that address the suggested mechanistic speculations, under the subsection “Possible synaptic bases of spatiotemporal receptive field reorganization”.

Reviewer #3:[…] My comments should be seen more as suggestions to improve an already strong paper rather than criticisms. Most of my comments relate with the interpretation of the results and are relatively minor.1) Abstract. The Abstract motivates the work by offering two alternative mechanisms of wiring development, elimination of redundant connections or formation of new connections. However, the general reader may think that this motivation is misleading since the development of neuronal connections frequently involves a combination of both. For example, each immature LGN neuron receives multiple retinal inputs that are eliminated as the brain matures. In addition, the remaining connections become stronger by making new connections (i.e. new synapses).2) Abstract. The parallel that the Abstract makes between elimination/formation of connections and input/intrinsic changes does not work well. The mixed mechanism involving changes in input connectivity and intrinsic excitability is different from a mixed mechanism involving elimination and formation of connections. The authors may want to consider changing the motivation of the work (e.g. nobody knew the intracellular changes underlying the development of direction selectivity until the authors performed these experiments).3) Abstract. It is unclear whether the authors can interpret the changes that they observe in the spatiotemporal receptive field as formation of new connections, as they claim in the Abstract. Changes in the temporal dimension of the spatiotemporal receptive field could be caused by combined changes in the receptive field size, time course and synaptic strength of already-connected inputs. A paragraph of the Discussion states this wonderfully (quoted below) but the rest of the paper (including the Abstract) gives the impression that the results provide a direct demonstration of changes in the number of inputs during development. This could be misleading particularly for the non-specialized reader. Quote of paragraph: "Alternatively, maturation of receptive field properties of already-connected LGN neurons or cortical neurons might cause the spatiotemporal receptive field to expand, and confer direction selectivity even without such selectivity in the antecedent inputs. […] Or, perhaps connections from LGN cells with reduced latencies do not form until after the onset of experience".4) Introduction and results. Several sections of the paper appear to imply that the results provide direct evidence for formation and elimination of synaptic inputs. However, as the authors acknowledge in the Discussion, this is one of several possible interpretations. Therefore, it is probably a good idea to make it clear early in the paper that the elimination/formation of inputs is not a direct conclusion from the results but a possible interpretation. For example, the authors state: "…became more elongated along the temporal axis, corroborating the idea of recruitment of new synaptic inputs. However, the subunits also became significantly narrower along their preferred space-time axis, consistent with pruning of synaptic inputs with certain spatiotemporal properties". The term "consistent with" is probably more appropriate than "corroborating". "Corroborating" may lead the non-specialized reader to believe that the authors are providing a direct demonstration of input recruitment.

We appreciate this concern, and, after consideration, we have changed the Introduction and Abstract substantially to be more agnostic as to the mechanisms that underlie the changes in receptive fields. We take the point that we do not have any direct evidence of synaptic changes and that the better thing to do is to focus on the receptive field changes we observed and describe the possible circuit mechanisms in the Discussion.

5) Changes in the relative strength of intracortical inhibition could be proposed as another possible interpretation of the results and should probably be discussed. Several studies have demonstrated that intracortical inhibition plays an important role in cortical direction selectivity including those from Sacha Nelson (the PhD advisor of the corresponding author) and, more recently, David Fitzpatrick (the postdoctoral advisor). An alternative interpretation of the results is that the receptive field maturation that they observe originates from changes in the relative strength of excitation and inhibition. For example, the much larger increase in spiking than Vm response during development could reflect a release from shunting inhibition as cortical neurons mature. This mechanism could also explain why the spiking response in the null direction is weaker in naive than mature ferrets. The developmental increase in response to the null direction may indicate that stimulus-driven excitation becomes stronger than stimulus-driven inhibition during development. Inhibition could also be relatively stronger in immature brains because cortical neurons spike less but each spike from inhibitory neurons has a relatively bigger impact in the cortical response because excitation is weak.

We have reorganized the Discussion to place the statements about the descriptive changes in spatiotemporal receptive structure (some expansion and some constriction) in one section, and speculation about the possible synaptic bases that might underlie these changes in another section. We appreciate the suggestion to separate these portions and expand the ideas of the synaptic bases to include inhibition. In the new section that speculates on synaptic bases, we consider 3 main themes: there could be recruitment of additional LGN cells; there could be no change in the LGN cell connections but the receptive field properties of the connected LGN cells change; there could be changes in the excitatory or inhibitory connections within cortex.

6) It may be a good idea to report the signal to noise ratio of the receptive field measurements. This is important to fully disregard the possibility that the reported differences are not partly due to differences in signal to noise between naïve and experienced animals. The signal to noise should be lower in naïve animals because stimuli drive weaker fluctuations in Vm and spike rate than in experienced animals. This is not a concern in the interpretation of the results because the authors did not find differences in area, spatial extent and temporal extent of the receptive fields between naïve and experienced animals. However, reporting signal to noise measurements somewhere in the Materials and methods section is important to fully eliminate this concern. If there is a difference in signal to noise, the authors can control for a possible contribution of this difference in their comparison by subsampling spikes in experienced animals at the same rate as in naïve animals and show that the most important results (e.g. changes in eccentricity) hold.

As the reviewer points out, since not all STRF shape parameters show differences, we think it is unlikely that SNR is an underlying confound here. Nevertheless, we took another look at the data to address this point. There is some uncertainty as to what would be the best estimate of SNR in this scenario, but one idea is that the actual correlation values that build the STRFs might contain some clues. The concern as stated by the reviewer is that the Vm responses are weaker in the naïve animals, and therefore might lead to unreliable STRF structures, which might lead to spurious differences in shape parameters. If this were the case, both the ON and OFF subunits in the naïve state would be less clearly separated from the correlation floor (noise or background). However, as can be seen from the example STRFs in Figure 3 (notice the color bars), there doesn’t appear to be a strong difference between the 2 groups in how well the STRFs are separated from the floor. To quantify, we ran a one-way ANOVA between the *absolute* peak correlation values of the ON and OFF subunits between the 2 groups, and found no significant differences. This demonstrates that the STRF subunit peaks and troughs are no differently separated from the floor in the 2 groups, and therefore this factor is unlikely to contribute to the observed shape differences (Author response image 1). A paragraph on this analysis has been added to the Materials and methods section.

7) Results. The following statement is another example of a possible misleading conclusion: "…In sum, following visual experience, the Vm receptive fields exhibited marked reorganization… and recruitment of lower latency inputs". "recruitment of lower latency inputs" seems to imply that the results directly show this. However, recruitment of lower latency inputs is one of several possible interpretations. One interpretation is that already-connected LGN inputs become stronger and faster in response latency. It would be very interesting to see the relation between response strength and latency across the entire dataset. One possibility is that response strength and latency are negatively correlated and both change together during development.

We agree with the reviewer’s point. In accordance with our complete revamping of the text to remove all allusion to the idea that the data unambiguously demonstrate a cortical origin of the STRF changes, we have removed two mentions of the “recruitment” idea from the Results section. Instead, all these ideas are elaborated as possible mechanisms in the Discussion section under the subsection “Possible synaptic bases of spatiotemporal receptive field reorganization”.

8) It is very impressive to see all the pronounced and clear changes that the authors report in just 14 days of development. Great work!

Thank you!

Some references that should probably be cited:Lien and Scanziani, 2018. To my knowledge, this is the only study measuring directly the thalamic inputs to directional selective cortical cells. It shows that, at least for some cortical cells, direction selectivity can be explained simply by the addition of thalamic inputs with different response time courses.Kremkow et al., 2016. This study shows that rapid changes in direction preference within 100-200 microns of horizontal cortical distance are closely associated with rapid changes in receptive field position. This finding is also consistent with the notion that changes in direction preference are associated with changes in the receptive field position of thalamic inputs.Wilson, Scholl and Fitzpatrick, 2018. There are many papers showing the importance of intracortical inhibition in generating cortical direction selectivity. Wilson et al., 2018, should probably be cited because it is the most recent one on this topic and uses ferrets, the animal model chosen by the authors.

Yes, thank you, all added in the Discussion.